

# Modelling long-term impacts of mountain pine beetle outbreaks on merchantable biomass, ecosystem carbon, albedo, and radiative forcing

Jean-Sébastien Landry[1,6], Lael Parrott[2], David T. Price[3], Navin Ramankutty[4], and H. Damon Matthews[5]

[1]Department of Geography, McGill University, Montréal, Canada
[2]Earth and Environmental Sciences and Biology, Irving K. Barber School of Arts and Sciences, University of British Columbia, Kelowna, Canada
[3]Natural Resources Canada, Canadian Forest Service, Northern Forestry Centre, Edmonton, Canada
[4]Liu Institute for Global Issues and Institute for Resources, Environment, and Sustainability, University of British Columbia, Vancouver, Canada
[5]Department of Geography, Planning and Environment, Concordia University, Montréal, Canada
[6]Currently at the Department of Geography, Planning and Environment, Concordia University, Montréal, Canada

*Correspondence to:* Jean-Sébastien Landry (jean-sebastien.landry2@mail.mcgill.ca)

**Abstract.** The ongoing major outbreak of mountain pine beetle (MPB) in forests of western North America has led to considerable research efforts. Yet many questions remain unaddressed regarding its long-term impacts, especially when accounting for the range of possible responses from the non-target vegetation (i.e., deciduous trees and lower-canopy shrubs and grasses). We used the Integrated BIosphere Simulator (IBIS) process-based ecosystem model along with the recently incorporated Marauding Insect Module (MIM) to quantify, over 240 years, the impacts of various MPB outbreak regimes on lodgepole pine merchantable biomass, ecosystem carbon, surface albedo, and the net radiative forcing on global climate caused by the changes in ecosystem carbon and albedo. We performed simulations for three locations in British Columbia, Canada, having different climatic conditions, and four scenarios of various coexisting vegetation types with variable growth release responses. The impacts of MPB outbreaks on merchantable biomass (decrease) and surface albedo (increase) were similar across the 12 combinations of locations and vegetation coexistence scenarios. The main finding from our study was that the impacts on ecosystem carbon and radiative forcing, on the contrary, varied substantially in magnitude and sign depending upon the presence and response of the non-target vegetation, particularly for the two locations not subjected to growing-season soil moisture stress. Despite major uncertainty in the value of the resulting radiative forcing, a simple analysis also suggested a smaller impact on global temperature from the MPB outbreak in British Columbia compared to one month of global anthropogenic $CO_2$ emissions from fossil fuel combustion and cement production. Moreover, we found that: (1) outbreak severity (i.e., per-event mortality) had a stronger effect than outbreak return interval on the variables studied, (2) MPB-induced changes in carbon dynamics had a stronger effect than concurrent changes in albedo on net radiative forcing, and (3) the physical presence of MPB-killed dead standing trees was potentially beneficial to tree regrowth.



## 1 Introduction

The mountain pine beetle (MPB; *Dendroctonus ponderosae* Hopkins) is an insect native to forests of western North America, from northern Mexico to British Columbia, Canada (Safranyik and Carroll, 2006). Outbreaks of this bark beetle are characterized by high stand-level mortality of the target species, primarily lodgepole pine (*Pinus contorta* var. *latifolia*), but also other pines and, occasionally, other genera (NRCan, 2012). The MPB outbreak that started at the end of the previous century has reached an unprecedented level of documented severity, particularly in British Columbia where 18.1 Mha of forests have been affected (British Columbia, 2012b) and more than half of the merchantable pine volume has been killed (Walton, 2013).

Forests generally appear to recover well following MPB outbreaks (Axelson et al., 2009; Kashian et al., 2011; Hansen, 2014; Alfaro et al., 2015), which have recurred in western North America for thousands of years (Brunelle et al., 2008). However, forest managers face the decision on whether to proceed with salvage logging of MPB-killed dead standing trees (DSTs) and how best to do it (Griesbauer and Green, 2006; Bowler et al., 2012; Amoroso et al., 2013; Hawkins et al., 2013; Mathys et al., 2013; Landry and Ramankutty, 2015). MPB impacts also go beyond timber losses by modifying ecosystem carbon storage, thereby possibly affecting the ongoing climate change. The recent MPB outbreak has been estimated to decrease ecosystem carbon storage (cumulative values) by 270 Tg C between 2000 and 2020 in British Columbia (Kurz et al., 2008), by 580 Tg C between 1999 and 2050 in British Columbia (Arora et al., 2016), and by 15–26 Tg C between 2000 and 2009 in the western United States (Ghimire et al., 2015), the last value increasing to 19–35 Tg C when including mortality from other bark beetles.

Recent reviews have identified various lingering knowledge gaps limiting the understanding of ecological and climatic effects caused by outbreaks of MPB and other forest insects (Liu et al., 2011; Seidl et al., 2011; Hicke et al., 2012a; Landry and Ramankutty, 2015). First, the mortality of many large trees often causes a growth release of the surviving non-target species and smaller host trees generally avoided by the MPB, which can alter the competition balance among the plant types present (Romme et al., 1986; Heath and Alfaro, 1990; Stone and Wolfe, 1996; Axelson et al., 2009; Amoroso et al., 2013; Hawkins et al., 2013; Hansen, 2014; Alfaro et al., 2015; Campbell and Antos, 2015). This growth release likely explains why field-based studies using the eddy covariance method have found the forest carbon balance to be more resilient than expected during an MPB outbreak or for close to a decade afterwards (Bowler et al., 2012; Brown et al., 2012; Reed et al., 2014). Therefore, modelling studies should allow for the possibility of a compensatory response from the surviving vegetation, including lower-canopy shrubs and grasses. Second, there is a need for more studies assessing the range of responses to different outbreak mortality levels and return intervals under the same background conditions, because comparisons performed across forest types and climates can be misleading. Third, the recurrence of MPB outbreaks calls for a long-term perspective going beyond a single mortality event. Fourth, the impact of the physical presence of MPB-killed DSTs on local exchanges of energy and water as well as the influence of these modified exchanges on carbon cycling have hitherto not been studied. Fifth, the climatic effects of MPB outbreaks are not limited to the carbon cycle, as the post-outbreak fall of DST needles and stems increases the reflection of incoming solar radiation, especially over seasonally snow-covered forests (Bright et al., 2013; Vanderhoof et al., 2014). In the only study to date aiming to quantify the net global climatic impact of MPB outbreaks, this albedo-induced cooling was estimated to be stronger than the warming from reduced ecosystem carbon storage (O'Halloran et al., 2012).



The main objective of our study was to use a modelling approach to evaluate the impacts of MPB outbreaks on four variables relevant to the forestry sector, land–atmosphere exchanges of carbon and energy, and global climate change, while explicitly addressing the five knowledge gaps identified above. While more uncertain than empirical studies, process-based modelling approaches can provide a longer-term perspective on the impacts of MPB outbreaks and help assess interactions among several

factors. Our purpose was not to forecast stand-level forest attributes (e.g., species-level basal area), but to contrast responses for very different scenarios about the presence and response of non-target vegetation. Similarly, we did not account for all the factors affecting MPB population dynamics because we imposed idealized outbreak regimes, seeking here to provide initial insights on how impacts varied as a function of outbreak severity and return interval (e.g., Dietze and Matthes, 2014).

## 2  Methods

### 2.1  Overview

Our approach involved a set of different scenarios of coexistence between the MPB-targeted trees and non-target vegetation types; for each of these scenarios, we compared, over 240 years and for three locations in British Columbia, the impacts of various MPB outbreak regimes. In each instance, we included the explicit representation of interactions between MPB-killed DSTs and the carbon, energy, and water cycles.

### 2.2  Modelling the effects of MPB mortality

We used the recently developed Marauding Insect Module (MIM) incorporated within the Integrated BIosphere Simulator (IBIS) to simulate the effects of insect outbreaks. Here, we provide an overview of IBIS–MIM and refer readers to Landry et al. (2016) for more details.

The IBIS global ecosystem model was originally developed to estimate, within a single and consistent modelling framework,

the land–atmosphere exchanges of carbon, energy, water, and momentum required by climate models, while simulating how vegetation phenology and spatial distribution respond to climate (Foley et al., 1996; Kucharik et al., 2000). IBIS represents coexisting upper (trees) and lower (shrubs and grasses) vegetation canopies as well as various soil and snow layers. The simulated exchanges of carbon, energy, and water depend upon the state of both canopies and the soil, including snow when present. Vegetation diversity is represented through various plant functional types (PFTs) characterized by different climatic

constraints and parameters related to physiology, carbon dynamics, and energy exchanges. Photosynthesis and autotrophic respiration are typically computed on an hourly time step as a function of input climatic conditions. Competition balance and vegetation changes are determined at the end of each year based on the annual carbon balance of each PFT, except for the leaf phenology of deciduous PFTs, which is updated daily. For each PFT that can exist in the grid cell based on prevailing climatic conditions, leaf area index (LAI) cannot become lower than a very small, but non-zero, value; if a PFT is entirely

removed, this seed LAI can therefore initiate regeneration. Annual litterfall is divided into daily transfers to soil, where carbon decomposition is modelled as a function of microbial biomass, soil temperature, and moisture. IBIS results compare relatively



well with empirical data over large regions and several field sites, including in Canada (Foley et al., 1996; Delire and Foley, 1999; Kucharik et al., 2000; Lenters et al., 2000; El Maayar et al., 2001, 2002; Kucharik et al., 2006).

MIM was designed to simulate the effects of insect outbreaks within process-based ecosystem models similar to IBIS (Landry et al., 2016). MIM prescribes, at a daily time step, the direct insect-caused vegetation damage (i.e., defoliation and/or

mortality), an approach that is similar to the "pathogen and insect pathways" from Dietze and Matthes (2014). The resulting impacts on vegetation dynamics and land–atmosphere exchanges of carbon, energy, and water are estimated by the supporting ecosystem model as a function of the post-outbreak state of the vegetation. MIM currently represents the effects of vegetation damage caused by outbreaks of three insect functional types (IFTs): broadleaf defoliators, needleleaf defoliators, and bark beetles. The bark beetle IFT used here was parameterized based on MPB-caused mortality of lodgepole pine. When a MPB

outbreak occurs, mortality is assumed to begin on 1 August and increases linearly over 50 days (Landry et al., 2016) until reaching the user-prescribed annual mortality level for the specific year (see Section 2.3). Killed trees become DSTs that interact with the exchanges of energy and water (e.g., absorbing solar radiation) but do not transpire or perform photosynthesis. DST carbon is gradually transferred to litter based on a pre-defined schedule: at year end for fine roots, over the three years following the year of mortality for needles, and, after a 5-year delay period, over 20 years for stem and coarse roots (Landry

et al., 2016). IBIS then subdivides these annual amounts into daily transfers to soil.

IBIS–MIM results after a simulated MPB outbreak generally agreed with previous field-, satellite-, and model-based studies for 28 variables related to vegetation dynamics and the exchanges of carbon, energy, and water, over daily to multi-annual timescales (Landry et al., 2016). The only bias identified in that previous assessment of IBIS–MIM consisted of a lower snow depth/amount following MPB mortality vs. the no-outbreak control, contrary to the conclusion of most – but not all – previous

studies. This bias likely resulted from overestimation by IBIS of heat storage within tree stems (Pollard and Thompson, 1995; El Maayar et al., 2001), including DSTs. IBIS–MIM might therefore underestimate the length of the snow cover season in MPB-attacked stands, thereby underestimating the consequent increases in springtime albedo and reflected solar radiation. This possible bias seems unlikely to be serious for the current study, because: (1) areal snow coverage, which matters more for albedo than snow depth/amount, was the same for the outbreak and control results during most of the snow cover season;

and (2) the generally earlier and faster snowmelt caused by MPB outbreaks (Pugh and Small, 2012; Mikkelson et al., 2013) is consistent with IBIS–MIM results.

### 2.3  Simulation design

IBIS requires input data for soil texture and climatic variables related to temperature, humidity (including precipitation and cloud cover), and wind speed. For all variables we used the same input data as Landry et al. (2016). Note that contrary to gap

models, IBIS does not have an intrinsic spatial resolution; but as the computation of radiative forcing requires a specific area (see Appendix A), we used a nominal area of 1 ha here. Although we did not assess the effect of climate change, seeking to first understand the ecosystem responses within a stable climate regime, we considered the effect of varying climatic conditions by studying three locations in British Columbia, henceforth designated as northern, central, and southern (Fig. 1). These three locations have witnessed substantial MPB mortality since 2000 (Walton, 2013) under different climatic conditions (Table 1).





The southern location is warmer than the central and northern locations. These last two locations have similar mean annual temperature, but the northern location has warmer summers and colder winters. Annual precipitation is similar in all locations, but summer rainfall is much lower in the southern location and results in drier conditions during the growing season.

We divided the simulations into four groups of different vegetation coexistence scenarios (Table 2). Five IBIS PFTs can coexist at the three locations considered here: the needleleaf evergreen (NE) trees targeted by MPB, broadleaf deciduous (BD) trees (e.g., trembling aspen; *Populus tremuloides* Michx.), and evergreen shrubs, cold-deciduous shrubs, and $C_3$ grasses in the lower canopy. The NEonly scenario allowed only NE trees and thus did not account for the possible response of the non-target vegetation. The NE-LC scenario allowed for the coexistence of NE trees and lower-canopy PFTs, but excluded BD trees. Note that in IBIS, coexisting PFTs compete for the same incoming solar radiation and soil water, instead of being segregated into independent sub-grid tiles as in many similar models, so that tree PFTs actually shade the underlying lower canopy. The NE-LCcons scenario was similar to NE-LC, except that the total biomass of the lower canopy was kept constant from the year of the first MPB outbreak onwards. Consequently, the lower canopy could increase its net primary productivity (NPP, in $\mathrm{kg\,C\,m^{-2}\,yr^{-1}}$) following MPB mortality (e.g., due to higher light penetration), but the additional productivity was transferred to litterfall so that the lower canopy was unable to grow and expand, thereby preventing further increases in productivity. The purpose of this constraint was to account for the effect of vegetation growth limitations not included in IBIS, such as nutrient availability. Finally, the AllPFT scenario allowed the five PFTs to freely compete throughout all years.

We performed 12 sets (three locations and four coexistence scenarios) of five independent simulations; note that all simulations at a given location had the exact same weather. For a given set, the five independent simulations branched from the same 400-year spin up and consisted of: (1) a no-outbreak *Control* run; (2) a single 100 % MPB mortality event occurring on year 1 after the spin up, used to assess the theoretical maximum impacts and designated the *Peak* regime; (3) a regime of periodic single-year MPB outbreaks, also starting on year 1, with a mortality level of $16.\overline{6}\,\%$ at return intervals of 20 years, designated the *Small* regime; (4) similar to *Small*, but with a single-year mortality level of $33.\overline{3}\,\%$ at return intervals of 40 years, designated the *Medium* regime; and (5) similar to *Small* and *Medium*, but with a single-year mortality level of 50.0 % at return intervals of 60 years, designated the *Large* regime. We simulated a spatially homogeneous distribution of MPB-killed trees, based on the observation that MPB mortality was spatially more regular than the underlying distribution of trees in a 0.25-ha plot (Kashian et al., 2011) and to avoid the complications of introducing sub-grid spatial heterogeneity in IBIS. All simulations were continued for 240 years after the spin up, for a total of 640 years. Note that over these last 240 years, the mean mortality was $0.8\overline{3}\,\%\,\mathrm{yr^{-1}}$ for the three periodic regimes, thereby allowing the effect of outbreak severity vs. return interval to be compared for the same mean mortality, but was only $0.41\overline{6}\,\%\,\mathrm{yr^{-1}}$ for the *Peak* regime. We simulated single-year mortality events instead of many-year outbreaks for two reasons. First, we wanted to focus on long-term results and a previous study with a model similar to IBIS found that, for the same level of total mortality occurring over 1, 5, or 15 years, differences in net ecosystem productivity (NEP, in $\mathrm{kg\,C\,m^{-2}\,yr^{-1}}$) became very small 25 years after the outbreaks (Edburg et al., 2011). Second, simulating a multi-year outbreak would raise the issues of its length and precise unfolding over consecutive years, introducing other complicating factors to our experimental setup that already considers the combination of three locations, four coexistence scenarios, and four outbreak regimes.



## 2.4 Response variables

We studied four variables: lodgepole pine merchantable biomass ($B_{merch}$, in $kg\,C\,m^{-2}$), ecosystem carbon ($C_{eco}$, in $kg\,C\,m^{-2}$), surface albedo ($\alpha$, unitless), and radiative forcing ($RF$, in $W\,m^{-2}$). $B_{merch}$ is highly relevant for the forestry sector, as it indicates the amount of lodgepole pine having commercial value. Note that, in the AllPFT scenario, BD trees (e.g., trembling aspen) could also have some commercial value, but we intentionally limited $B_{merch}$ to lodgepole pine due to the major commercial importance of this species in British Columbia. $C_{eco}$ goes beyond timber and accounts for all the carbon contained in live and dead pools including the soil, so that changes in $C_{eco}$ directly affect atmospheric $CO_2$ levels. Changes in $\alpha$ affect energy exchanges, with increases in $\alpha$ implying a cooling influence on the global climate. Finally, $RF$ is used to assess the net impact of different perturbations on the global mean atmospheric surface temperature, without performing simulations with climate models (Myhre et al., 2013). In this study, the net $RF$ (indicating warming if $RF > 0$, cooling if $RF < 0$) following MPB outbreaks was the sum of the radiative forcing from changes in atmospheric $CO_2$ and $\alpha$. Unlike the other variables, $RF$ is defined as a change between two different states; hence, $RF$ results cannot be provided on a relative change (%) basis but must be absolute values for a given outbreak area. We stress that even if MPB impacts were simulated for a nominal area of 1 ha, the $RF$ results we report are, by definition, for the net effect on global climate. We explain in Appendix A how we used IBIS output to compute $B_{merch}$ and $RF$.

## 2.5 Simplified analysis of maximum global climatic impact

Our simulation design was too simple to allow for a precise estimate of the global climatic impact from the MPB outbreak in British Columbia, but we used our $RF$ results to bound the maximum value of this net warming or cooling impact. To do so, we identified among all our $Peak$ simulations the two instances that led to the most extreme (positive and negative) annual $RF$ values. We then recomputed these $RF$ trajectories for an area of 18.1 Mha, which is the total area affected by the MPB outbreak (British Columbia, 2012b). Finally, we determined the value of a single pulse of actual (positive $RF$) or avoided (negative $RF$) fossil fuel $CO_2$ emissions that would invariably have, throughout the 240 years, a stronger radiative forcing than the MPB-caused $RF$ (see Appendix A). This approach likely overestimated the maximum annual impact of MPB on the global climate for three reasons. First, not all area affected by MPB suffered 100 % mortality as prescribed in $Peak$. Second, the single-year $Peak$ mortality led to stronger extreme $RF$ values compared to a gradual increase and decrease of the outbreak over many years. Third, the amount and composition of non-target vegetation are highly variable among MPB-attacked stands (Axelson et al., 2009; Amoroso et al., 2013; Hawkins et al., 2013; Pelz and Smith, 2013; Alfaro et al., 2015; Campbell and Antos, 2015) and this variability appears consequential for $RF$ (see Section 3); hence the $RF$ response from the same vegetation coexistence scenario was unlikely representative of the mean response over the entire area affected.



## 3  Results

### 3.1  Transient results

MPB-caused reductions in $B_{merch}$ were similar across the three locations and four vegetation coexistence scenarios (Fig. 2).
For the $Peak$ regime, the single 100 % mortality event removed all $B_{merch}$, after which 20–50 years were needed before NE
trees became big enough to have any commercial value. The slower recovery of $B_{merch}$ in AllPFT compared to other scenarios
resulted from the growth release of BD trees, which were able to grow strongly for a few decades but were very poor long-term
competitors at all locations. For the three periodic regimes, the recurring MPB outbreaks prevented $B_{merch}$ from recovering
to the $Control$ value as in $Peak$. We found some evidence of a biophysical interaction between DSTs and regrowing NE
trees for the $Peak$ regime in the NEonly, NE-LCcons, and NE-LC scenarios: after the 100 % mortality event, NPP of NE
trees increased rapidly but, after about 20–25 years, decreased noticeably for ∼10 years before resuming again (similarly to
the results shown in Fig. 2 from Landry et al., 2016). We believe this response resulted from the interception of radiation by
DSTs, which warmed the upper canopy and initially allowed regrowing NE trees to perform photosynthesis more efficiently at
a higher temperature. This warming effect decreased as DSTs fell, causing productivity of the regrowing NE trees to decline
even though they were intercepting more light. In the AllPFT scenario, the slower regrowth of NE trees caused them to miss
the warming effect due to the presence of DSTs.

The impacts of MPB on $C_{eco}$ (Fig. 3) were much more complex than on $B_{merch}$. In the NEonly scenario, the three periodic
regimes led to gradual declines in $C_{eco}$ that showed indications of possible stabilization towards the end of the simulations,
whereas for the $Peak$ regime, $C_{eco}$ was still recovering after 240 years. The results were qualitatively the same in the NE-
LCcons scenario, albeit with much smaller $\Delta C_{eco}$ because the lower-canopy growth release partially compensated for the death
of NE trees. At the southern location, the growing-season soil moisture stress probably explains why the growth release of non-
target PFTs was only marginally stronger in the unconstrained NE-LC and AllPFT scenarios vs. NE-LCcons. Conversely, MPB
outbreaks substantially increased total NPP at the northern and central locations for NE-LC and AllPFT, by inducing a strong
growth release of the non-target vegetation and fostering the increased coexistence of PFTs occupying different ecological
niches (upper vs. lower canopies, and evergreen needleleaf vs. deciduous broadleaf strategies) compared to undisturbed forests
dominated by lodgepole pine. Here, the higher productivity of the non-target vegetation exceeded the productivity losses and
gradual decomposition of killed NE trees; hence, after a delay of a few years to a few decades, $\Delta C_{eco}$ switched from negative
to positive (Fig. 3, panels g, h, j, and k).

MPB outbreaks increased $\alpha$ for all locations and vegetation coexistence scenarios (Fig. 4). Results were very similar across
locations and scenarios, except for smaller $\alpha$ increases in AllPFT (panels j, k, and l) for the $Peak$ regime due to the absorption
of radiation by BD trees (even when leafless during winter, as IBIS accounts for the snow-masking effect of stems) following
their growth release.

$RF$ varied substantially across locations and vegetation coexistence scenarios (Fig. 5). For NEonly and NE-LCcons, and
for the other scenarios at the southern location only, $RF$ was almost always positive, indicating a warming effect of MPB
outbreaks on the global climate. The small negative $RF$ values observed for the $Peak$ regime in three instances (panels d, e,




and l) came from a combination of still slightly positive $\Delta\alpha$, almost complete return of $C_{eco}$ to the $Control$ values (see Fig. 3), and long time lags in atmosphere–ocean $CO_2$ exchanges following vegetation regrowth (see Landry and Matthews (2016) for such time lags after the simulation of fire disturbances in a coupled climate–carbon model; here, these time lags were accounted for by the convolution approach used to compute $RF$ as explained in Appendix A). At the northern and central locations, MPB

outbreaks under the NE-LC and AllPFT scenarios led instead to a net cooling effect on the global climate, even though $RF$ values were initially positive for at least four years (panels g, h, j, and k).

### 3.2 Mean effect

Figure 6 shows the mean time-averaged effect of MPB outbreaks on the four variables over the 240 years following the first outbreak. In the NEonly scenario, the results were almost equal in all three locations for a given MPB regime despite the

different climatic conditions. When other PFTs coexisted with the target NE trees, however, the influence of climate became noticeable, especially in the NE-LC and AllPFT scenarios. Figure 6 also reveals that a higher per-event mortality generally caused stronger absolute effects than a shorter return interval. Indeed, for a given location and vegetation coexistence scenario, the departure from zero for the periodic outbreak regimes was generally in the following order: $Large > Medium > Small$. Moreover, the single 100 % mortality event under the $Peak$ regime had a mean effect comparable to, and in many instances

greater than, the mean effect under the three periodic regimes, despite a 240-year mean mortality of $0.41\overline{6}$ % yr$^{-1}$ for $Peak$ vs. $0.8\overline{3}$ % yr$^{-1}$ for the periodic regimes.

Figure 6 also suggests that $RF$ was more closely linked to $\Delta C_{eco}$ than to $\Delta\alpha$, which is supported by the transient results where $RF$ basically mirrored $\Delta C_{eco}$ (compare Figs. 3 and 5). Even though the $\alpha$-caused cooling effect offset a fraction of the $CO_2$-based warming when $\Delta C_{eco}$ was negative or added to the $CO_2$-based cooling when $\Delta C_{eco}$ was positive, the sign

of $RF$ was primarily related to $C_{eco}$ changes. Spearman's rank correlation coefficient between $\Delta C_{eco}$ and $RF$ across the 48 outbreak simulations presented in Fig. 6 was $-0.99$, indicating that greater decreases (increases) in $C_{eco}$ were almost invariably associated with greater increases (decreases) in $RF$, thereby leading to a greater warming (cooling) effect. On the contrary, Spearman's rank correlation coefficient between $\Delta\alpha$ and $RF$ was only $+0.21$, indicating a weak association that was opposite to the actual effect of increased $\alpha$ on $RF$.

### 3.3 Comparison of MPB to anthropogenic $CO_2$ emissions

The highest (positive) yearly $RF$ value came from the NEonly scenario at the central location (Fig. 5b), whereas the lowest (negative) yearly $RF$ value came from the NE-LC scenario at the northern location (Fig. 5g). We therefore recomputed these two $RF$ trajectories over an area of 18.1 Mha to obtain the bounding positive and negative responses, respectively. As shown in Fig. 7, these bounding $RF$ responses invariably had smaller absolute impacts than a single pulse of $+0.83$ Pg C of fossil fuel

$CO_2$ for the warming case, or a single pulse of $-0.80$ Pg C (avoided emissions) for the cooling case.





## 4 Discussion

### 4.1 Influence of the non-target vegetation

Our main finding is that the non-target vegetation has a major influence on ecosystem carbon storage and net climatic impact following MPB outbreaks. The substantial variability in $C_{eco}$ (Fig. 3) and $RF$ (Fig. 5) responses occurred despite almost iden-

tical recovery of the MPB-targeted dominant tree species across the four vegetation coexistence scenarios and three locations (Fig. 2). Previous studies not accounting for growth release of the non-target vegetation, including shrubs and grasses, may therefore have overestimated the MPB-caused decreases of $C_{eco}$. Strong compensatory responses following MPB or other bark beetle outbreaks have also been reported in previous studies that used controls or considered several mortality levels (Heath and Alfaro, 1990; Stone and Wolfe, 1996; Klutsch et al., 2009; Griffin et al., 2011; Amoroso et al., 2013; Hawkins et al., 2013),

including cases of NPP or aboveground tree biomass being higher with than without insect outbreaks for some period of time (Romme et al., 1986; Seidl et al., 2008; Pfeifer et al., 2011; Hansen, 2014). Modelling studies on a centennial timescale also found that $C_{eco}$ can be higher when insect disturbances are simulated than without them (Seidl et al., 2008; Albani et al., 2010), while field-based studies have concluded that the growth release of the surviving vegetation can contribute to a fast recovery of NEP following MPB outbreaks (Bowler et al., 2012; Brown et al., 2012) and even to stable growing-season NEP despite

ongoing increases in MPB mortality (Reed et al., 2014). Another field-based study found $C_{eco}$ to be almost equal in control stands and stands affected 25–30 years earlier by a $\sim$25 %-mortality MPB outbreak (Kashian et al., 2013).

We do not believe that any of the vegetation coexistence scenarios we simulated is fundamentally more realistic than the others. Rather, we believe that these four scenarios sample the ensemble of possible responses to MPB outbreaks, because the amount and composition of non-target vegetation vary substantially among MPB-attacked stands, even over short distances,

which leads to variable regeneration pathways (Axelson et al., 2009; Amoroso et al., 2013; Hawkins et al., 2013; Pelz and Smith, 2013; Alfaro et al., 2015; Campbell and Antos, 2015). Given this substantial variability, trying to bracket the range of possible vegetation responses, like we did here, appears safer for large-scale modelling studies than aiming to forecast one specific course of events. Contrary to our simulation protocol, however, it is unlikely that the composition of non-target vegetation would remain unchanged from one outbreak to the next. Moreover, most forests in western North America undergo recurrent

stand-clearing fires or wood harvests, resetting stands on trajectories that, according to the evidence currently available, are not strongly affected by the previous occurrence of MPB outbreaks (Hicke et al., 2012b; Harvey et al., 2014).

### 4.2 Climatic impact

The only previous study estimating the net impact of MPB outbreaks on global climate found a negative $RF$ throughout the 14-year period studied, due to an $\alpha$-based cooling that was invariably stronger than the $C_{eco}$-based warming (O'Halloran et al.,

2012). This outcome is at odds with IBIS–MIM results, because our net $RF$ depended critically upon the sign of the $C_{eco}$ change and, even for instances of overall net cooling, our $RF$ values were positive during the first four years at least. This discrepancy likely involve methodological differences between the two studies, but might also come from a spatiotemporal mismatch that could have affected the $RF$ results from O'Halloran et al. (2012): their $\Delta\alpha$ was representative of MPB-killed



stands (coming from time-since-mortality comparisons at the stand level), whereas their $\Delta C_{eco}$ was based on the regional-level results of Kurz et al. (2008), which were for stands killed at different times averaged with unaffected stands.

Our estimate of the global climatic impact due to the MPB outbreak in British Columbia (Fig. 7), although very simple, seems appropriate to bound the range of possible values. For the warming case, the maximum decrease in $C_{eco}$ (based on $Peak$ from Fig. 3b, over 18.1 Mha) was equal to 818 Tg C $\sim$50 years after mortality. By comparison, Kurz et al. (2008) simulated a 2000–2020 decrease of 270 Tg C with an inventory-based model omitting the possible growth release of the non-target vegetation and projections of MPB-caused mortality almost 40 % higher than more recent estimates (Walton, 2013), whereas Arora et al. (2016) simulated a 1999–2050 decrease of 580 Tg C with a vegetation coexistence scenario similar to NEonly in an IBIS-like model. For the cooling case, the lower-canopy growth release in the unconstrained NE-LC scenario was likely too strong, causing the increase in $C_{eco}$ to be overestimated. Consequently, the actual impact probably lies within these bounding responses, which have smaller absolute $RF$ than a single pulse of $+0.83$ Pg C (warming case) or $-0.80$ Pg C (cooling case) of fossil fuel $CO_2$. Pulses of such magnitude represent approximately one month of current global $CO_2$ emissions from fossil fuel combustion and cement production (Le Quéré et al., 2015). Even though these results suggest a marginal impact on global temperature, the current MPB outbreak in British Columbia could add or offset a sizable share of the warming due to greenhouse gas (GHG) emissions from the province alone or Canada as a whole. Total GHG emissions were estimated at 61.5 Mt $CO_2$eq in British Columbia for 2012 (British Columbia, 2012a) and at 726 Mt $CO_2$eq in Canada for 2013 (Environment Canada, 2014), which are roughly equivalent to 0.017 and 0.20 Pg C of fossil fuel $CO_2$, respectively. Therefore, the upper bound on the maximum global climatic impact of the current MPB outbreak in British Columbia, for either warming ($+0.83$ Pg C) or cooling ($-0.80$ Pg C), is equivalent to GHG emissions over roughly 50 years for British Columbia and 4 years for Canada. A more adequate assessment would require a dedicated study going beyond the simplified analysis presented here, which could only provide an upper bound for either a net warming or net cooling effect.

It is important to remember that the $RF$ concept does not apply to changes in local temperature. Since the $\alpha$-based cooling is local whereas the $CO_2$-based effect is global, one could expect that MPB outbreaks always decrease local temperature. However, this perspective neglects the post-MPB changes in sensible and latent heat fluxes that also modulate local temperature. In summer, high levels of MPB mortality have been found to increase surface temperature by up to a few °C due to reduced evapotranspiration (Griffin et al., 2011; Bright et al., 2013; Maness et al., 2013), a response Landry et al. (2016) obtained in their detailed assessment of IBIS–MIM. Landry et al. (2016) also found that MPB outbreaks decreased surface temperature in winter, but could not find any empirical observations on this variable.

### 4.3 Management and research implications

Management activities aiming to prevent or respond to MPB outbreaks must consider several factors, including economic impacts (e.g., Patriquin et al., 2005) and potential effects on fire behaviour (see the Hicke et al., 2012b, review). Although we did not account for these factors or model management activities explicitly, our study suggests the following. First, our results are in line with an emerging body of empirical literature pointing towards the resilience of carbon storage in MPB-affected forests, due to the growth release of the surviving vegetation (Bowler et al., 2012; Brown et al., 2012; Reed et al., 2014).



Similar to previous studies (Stone and Wolfe, 1996; Klutsch et al., 2009; Griffin et al., 2011; Bowler et al., 2012; Vanderhoof et al., 2014), we found a growth release of shrubs and grasses – and not only trees – that could contribute to ecosystem-level resilience in carbon storage. This benefit could be compromised if non-target vegetation is damaged during salvage logging operations, which might therefore be detrimental to carbon stewardship. We found indications of the potential growth release of surviving target trees in the fast NPP recovery of NE trees for the three periodic regimes, and in the smaller $B_{merch}$ mean decrease for the three periodic regimes vs. $Peak$ despite higher average mortality rates (Fig. 6). This last outcome is consistent with the relative stability of aboveground wood NPP for mortality levels below ∼60% in a forest-level manipulation experiment (Stuart-Haëntjens et al., 2015).

Second, our $Peak$ results indicate that high amounts of DSTs could facilitate the growth of the surviving trees by warming the surrounding air. Salvage logging would therefore prevent this beneficial impact (in cool growing seasons) and dampen the growth release of surviving trees. The warming effect of DSTs appears coherent with empirical evidence on vegetation–temperature interactions in northern latitudes (Liu et al., 2006; Lee et al., 2011), but is likely smaller than simulated here due to the overestimation by IBIS of heat storage within tree stems. The impact of DSTs on the amount and partitioning (i.e., direct versus diffuse) of solar radiation absorbed by the surviving vegetation, which all modulate NPP, is also probably more complex than simulated by IBIS–MIM. Field studies comparing the vertical profiles of temperature, total solar radiation, and diffuse solar radiation of stands with and without DSTs would be useful to resolve such questions.

Third, since MPB outbreaks do not necessarily warm the global climate, outbreak-preventing activities like pre-emptive logging might not mitigate climate change. Assessing the net climatic impact of salvage logging is even more complicated, because the exercise needs to go beyond comparing $RF$ for salvaged and unsalvaged stands, and account for the landscape-level redistribution of harvesting activities as well as the differences in the production and fate of the salvaged-derived wood products compared to the no-salvage baseline (Lemprière et al., 2013; Landry and Ramankutty, 2015).

Fourth, IBIS–MIM transient results suggest that MPB impacts vary substantially not only across space but also through time. While the spatial variability of vegetation responses (Griesbauer and Green, 2006; Axelson et al., 2009; Amoroso et al., 2013; Hawkins et al., 2013; Pelz and Smith, 2013; Alfaro et al., 2015; Campbell and Antos, 2015) underlines the need for studies in many stands, the temporal variability reported here calls for continued or periodic data gathering over decades at the same sites. A better appreciation of the long-term effects of MPB outbreaks should foster adequate management responses in affected forests within the insect's native range, and also in the Canadian boreal forest where the MPB has already established and may spread as the regional climate warms (Cullingham et al., 2011; Nealis and Cooke, 2014).

### 4.4 Study limitations

Eddy covariance measurements of ecosystem-level carbon exchanges following MPB outbreaks (Bowler et al., 2012; Brown et al., 2012; Reed et al., 2014) extend for less than a decade and lack formal controls, hence limiting our capacity to validate long-term IBIS–MIM output. Many dendrochronological studies have gathered data over longer time periods, sometimes with formal controls, but the results do not translate directly into changes in stand-level NPP. Romme et al. (1986) estimated the effect of MPB on aboveground tree growth, but only for six stands 10–20 years after an outbreak. A single study by Kashian




et al. (2013) quantified the impacts of MPB on $C_{eco}$, through allometric equations and soil samples obtained at 12 stands 25–30 years after mortality; however, this study dealt primarily with fire and the MPB results were qualified as "preliminary" by the authors. Other empirical and modelling studies accounting for possible growth releases support the plausibility of IBIS–MIM responses when simulating additional PFTs besides NE trees, yet our results must be considered tentative because

process-based modelling of vegetation competition (Kucharik et al., 2006; Moorcroft, 2006; Purves and Pacala, 2008) and non-stand-replacing disturbances (Bond-Lamberty et al., 2015) remains challenging.

We could not capture the effect of nitrogen availability on post-MPB vegetation recovery (Edburg et al., 2011), because IBIS does not simulate nutrient cycling. Another limitation of IBIS is the representation of a single NE tree PFT and a single BD tree PFT, whereas tree species other than lodgepole pine and trembling aspen often coexist in MPB-attacked forests (Griesbauer

and Green, 2006; Axelson et al., 2009; Amoroso et al., 2013; Hawkins et al., 2013; Campbell and Antos, 2015). Accounting for these other species could increase the range of post-MPB responses for $C_{eco}$ and $RF$, and also partly offset the $B_{merch}$ reductions simulated here, which included lodgepole pine only. Since IBIS does not represent different age cohorts within the same PFT, we could not formally assess the growth release of individual surviving NE trees. The post-MPB response of younger trees would likely differ from those of mature trees and MPB impacts on tree demographics could further complicate

stand-level responses following an outbreak, for example by increasing total biomass despite reduced productivity because of a strong decrease in competition-related mortality (Pfeifer et al., 2011). IBIS parameters for the different PFTs were also not specifically based on data gathered from British Columbian forests.

Finally, the overestimated heat storage in tree stems could underestimate the MPB-caused $\alpha$ increase, thereby biasing $RF$ towards a warming effect. Conversely, another potential bias could overestimate the cooling effect of $\alpha$ increase: while a

seminal study concluded that $\alpha$ and atmospheric $CO_2$ act with the same "efficacy" on the global surface temperature (Hansen et al., 2005), other studies have found that $\alpha$ has a lower efficacy than $CO_2$ (Hansen et al., 1997; Davin et al., 2007).

## 5   Conclusions

Despite major progress over the last decades, various knowledge gaps still limit the understanding of the consequences of mountain pine beetle (MPB) outbreaks. In this study, we used a process-based ecosystem model to estimate the impacts

of MPB outbreaks on four variables relevant to the forestry sector, land–atmosphere exchanges of carbon and energy, and global climate change, while explicitly accounting for: (1) different vegetation coexistence scenarios and strengths of the post-outbreak growth release, including lower-canopy shrubs and grasses (Table 2); (2) different outbreak severities and return intervals; (3) the long-term effect of repeated outbreaks; (4) the biophysical influence of MPB-killed dead standing trees (DSTs); and (5) the effect of changes in surface albedo ($\alpha$) on the net radiative forcing ($RF$) resulting from MPB outbreaks.

Using a climate-driven process-based model further allowed us to compare responses across three locations in British Columbia (Fig. 1) having different climatic conditions (Table 1).

We found that the differences in vegetation coexistence scenario and location had little influence over MPB impacts on lodgepole pine merchantable biomass ($B_{merch}$; Fig. 2) and $\alpha$ (Fig. 4). On the contrary, accounting for the non-target vegetation



invariably reduced losses of ecosystem carbon ($C_{eco}$) and, at the two locations not subjected to growing-season soil moisture stress, even led to post-outbreak $C_{eco}$ increases for the two vegetation coexistence scenarios with the strongest growth release following MPB mortality (Fig. 3). Although MPB-induced increases in $\alpha$ always had a cooling influence, the net global warming or cooling effect of MPB outbreaks was determined by the much stronger carbon-based responses (see Fig. 6, and

compare Fig. 5 with Fig. 3). A simple analysis suggested that the MPB outbreak in British Columbia will have less influence on global temperature over the coming centuries than one month of global anthropogenic $CO_2$ emissions at the 2014 level (Fig. 7). We also found that higher outbreak severity (i.e., per-event mortality) usually caused a stronger mean long-term effect than a shorter return interval between outbreaks (Fig. 6) and that, by warming the canopy as a result of radiation absorption, DSTs might increase the productivity of surviving and regrowing trees.

The management and research implications of our study are fourfold. First, salvage logging, particularly when performed as clear-cut harvesting, may be detrimental to carbon stewardship when surviving trees and other lower-canopy vegetation are abundant. Second, salvage logging could slow forest recovery if, following high MPB mortality, tree productivity is indeed increased due to the physical presence of DSTs, a hypothesis that should be subject to empirical studies. Third, MPB disturbances might not necessarily lead to global warming, so activities aiming to prevent or control outbreaks (e.g., pre-emptive logging)

should not be heralded as climate mitigation strategies without more detailed analyses. Fourth, the substantial spatiotemporal variability in MPB-induced changes suggests a need to support field studies that encompass a wide range of stand conditions and are maintained over several decades.

## Appendix A:  Additional computations

### A1    Merchantable biomass

$B_{merch}$ is a fraction of the total tree biomass in a forest, because immature trees, as well as tops branches, and stumps from mature trees, are excluded. We computed $B_{merch}$ as the product of $prop$, which is the proportion (unitless) of the total tree biomass that is merchantable, and $B_{tot}$, which is the total tree biomass (in $\mathrm{kg\,C\,m^{-2}}$) estimated by IBIS. We derived $prop$ from Figure 5 of Kurz et al. (2009), which shows $B_{merch}$ and $B_{tot}$ as a function of time for a lodgepole pine stand:

$$prop = \begin{cases} 0 & \text{if } B_{tot}/B_{max} < 0.21 \\[2ex] 0.5058 + 0.3172 \times \ln\left(B_{tot}/B_{max}\right) & \text{otherwise} \end{cases} \tag{A1}$$

where $B_{max}$ is the maximum tree biomass (in $\mathrm{kg\,C\,m^{-2}}$) at equilibrium. The logarithmic function for $prop$ provided a good fit ($R^2 = 0.996$) with the data extracted from Kurz et al. (2009).





## A2 Radiative forcing

### A2.1 Mountain pine beetle

The net $RF$ was the sum of the radiative forcing from atmospheric $CO_2$ changes ($RF_{CO_2}$) and from $\alpha$ changes ($RF_{alb}$). $RF_{CO_2}$ in year $y$ caused by a change in atmospheric $CO_2$ in the same year ($\Delta C(y)$, in ppmv) compared to a reference concentration

($C_o$, in ppmv) was given by (Myhre et al., 1998):

$$RF_{CO_2}(y) = 5.35 \times \ln\left(1 + \frac{\Delta C(y)}{C_o}\right) \tag{A2}$$

The change in atmospheric $CO_2$ in year $y$ due to MPB outbreaks resulted from all past changes in $C_{eco}$ computed by IBIS–MIM, including vegetation regrowth, while accounting for the airborne fraction of these past fluxes. In other words, $\Delta C(y)$ was the convolution of the series of past yearly land-to-atmosphere fluxes with the impulse response function ($IRF$) for the

airborne fraction of these past fluxes. Since $\Delta C(y)$ is an absolute amount and not a change per unit of land area disturbed, we must compute it for a specific area $A_{MPB}$ (in m$^2$) of MPB mortality. We thus computed $\Delta C(y)$ as:

$$\Delta C(y) = A_{MPB} \times k \times \sum_{t=0}^{y-1} \delta C_{eco}(y-t) \times IRF(t) \tag{A3}$$

where $k$ is equal to $4.69 \times 10^{-13}$ ppmv per kg C (CDIAC, 2012) and $\delta C_{eco}(x) = (C_{eco,control}(x) - C_{eco,control}(x-1)) - (C_{eco,MPB}(x) - C_{eco,MPB}(x-1))$. For the $IRF(t)$ function (unitless), we used the mean response from the injection of a

single pulse of $CO_2$ into 15 different coupled climate–carbon models (Joos et al., 2013). A similar approach has already been used to estimate $RF_{CO_2}$ from MPB outbreaks (O'Halloran et al., 2012).

We estimated the radiative forcing in year $y$ caused by a change in $\alpha$ as the mean of monthly values:

$$RF_{alb}(y) = \frac{1}{365} \times \sum_{m=1}^{12} n_{days}(m) \times RF_{alb}(m,y) \tag{A4}$$

where $n_{days}(m)$ is the number of days in month $m$, and $RF_{alb}(m,y)$ is the average $\alpha$-caused radiative forcing in month

$m$ of year $y$. To estimate $RF_{alb}(m,y)$, we used the radiative kernels approach (Shell et al., 2008; Soden et al., 2008) which has already been employed in previous studies on MPB-induced changes in $RF_{alb}$ (O'Halloran et al., 2012; Vanderhoof et al., 2014). Using the $\alpha$ radiative kernel for month $m$ ($K_{alb}(m)$, in W m$^{-2}$), we could thus estimate the $\alpha$-caused radiative forcing as (Shell et al., 2008; Soden et al., 2008):

$$RF_{alb}(m,y) = \frac{A_{MPB}}{A_{Earth}} \times K_{alb}(m) \times \Delta\alpha(m,y) \tag{A5}$$




where $A_{Earth}$ is the Earth area ($5.1 \times 10^{14}\,\mathrm{m}^2$; Wallace and Hobbs (2006)) and $\Delta\alpha(m,y)$ is the change in $\alpha$ between a simulation with MPB mortality and the $Control$ simulation. We averaged $K_{alb}(m)$ from two models: the Community Atmospheric Model (data downloaded from http://people.oregonstate.edu/~shellk/kernel.html) and the Geophysical Fluid Dynamics Laboratory atmospheric model (data downloaded from http://www.rsmas.miami.edu/personal/bsoden/data/kernels.html).

5    We computed the net $RF$ for an outbreak area of 1 ha (i.e., $A_{MPB} = 10{,}000\,\mathrm{m}^2$). Strictly speaking, the $RF$ results we obtained cannot be directly scaled as a function of the disturbed area, because Eq. (A2) is not linear and the $K_{alb}(m)$ values in Eq. (A5) vary spatially. Yet both these restrictions can be ignored for regional-level scaling of our $RF$ results. First, the changes in total atmospheric $CO_2$ were very small, so that Eq. (A2) varied almost linearly with $\Delta C(y)$ as $\ln(1 + x) \approx x$ for very small values of $x$. Second, $K_{alb}(m)$ values were computed at a coarse resolution of $\geq 2°$ and are spatially correlated. Therefore, $RF$

10   for a MPB-disturbed area of 10,000 ha, for example, would be very well approximated by multiplying our reported results by a factor of 10,000.

### A2.2   Fossil fuel $CO_2$

The radiative forcing caused by a positive or negative pulse of fossil fuel $CO_2$ emissions was also computed based on Eq. (A2), with $\Delta C(y)$ being given by:

$$\Delta C(y) = k \times P \times IRF(y - 1) \tag{A6}$$

where $P$ is the value of the single pulse of emissions (in kg C) occurring on year 1, with $k$ and $IRF$ as in Eq. (A3). We varied $P$ until obtaining a radiative forcing response that was invariably greater (smaller) than the bounding MPB-caused positive (negative) $RF$ response throughout the 240 years.

*Author contributions.* J.-S. Landry designed the study with advice from L. Parrott, D. T. Price, and N. Ramankutty; J.-S. Landry performed
20   the simulations with IBIS–MIM and analyzed the results; J.-S. Landry prepared the manuscript with contributions from all co-authors.

*Acknowledgements.* We thank Dany Plouffe for producing Fig. 1 and the Concordia Climate Lab for feedback on the results. J.-S. Landry
was funded by a doctoral scholarship (B2) from the Fonds de recherche du Québec – Nature et technologies (FRQNT).



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

**Figure 1.** Three locations studied; the province of British Columbia is shaded.





**Figure 2.** Transient effect of the different MPB outbreak regimes on lodgepole pine merchantable biomass ($B_{merch}$) compared with the no-outbreak *Control* run (first outbreak occurred on year 1). The columns correspond to the three locations (Fig. 1) and the rows to the four vegetation coexistence scenarios (Table 2). *Control* values differed among locations and vegetation coexistence scenarios, so the same relative change (in %) across the 12 panels does not correspond to the same absolute change.





**Figure 3.** Transient effect of the different MPB outbreak regimes on ecosystem carbon ($C_{eco}$) compared with the no-outbreak $Control$ run (first outbreak occurred on year 1). The columns correspond to the three locations (Fig. 1) and the rows to the four vegetation coexistence scenarios (Table 2); the y-axis scale differs across the four rows. $Control$ values differed among locations and vegetation coexistence scenarios, so the same relative change (in %) across the 12 panels does not correspond to the same absolute change.





**Figure 4.** Transient effect of the different MPB outbreak regimes on surface albedo ($\alpha$) compared with the no-outbreak *Control* run (first outbreak occurred on year 1). The columns correspond to the three locations (Fig. 1) and the rows to the four vegetation coexistence scenarios (Table 2). *Control* values differed among locations and vegetation coexistence scenarios, so the same relative change (in %) across the 12 panels does not correspond to the same absolute change.





**Figure 5.** Transient effect of the different MPB outbreak regimes on radiative forcing ($RF$; in pico-W m$^{-2}$, for 1-ha outbreaks) compared with the no-outbreak $Control$ (first outbreak occurred on year 1). The columns correspond to the three locations (Fig. 1) and the rows to the four vegetation coexistence scenarios (Table 2); the y-axis scale differs across the four rows.





**Figure 6.** Mean effect over 240 years of the different MPB outbreak regimes on lodgepole pine merchantable biomass ($B_{merch}$), ecosystem carbon ($C_{eco}$), surface albedo ($\alpha$), and radiative forcing ($RF$; in pico-W m$^{-2}$, for 1-ha outbreaks) compared with the no-outbreak $Control$, for the three locations (Fig. 1) and four vegetation coexistence scenarios (Table 2). $Control$ values differed among locations and vegetation coexistence scenarios, so the same relative change (in %; panels a through i) does not correspond to the same absolute change.





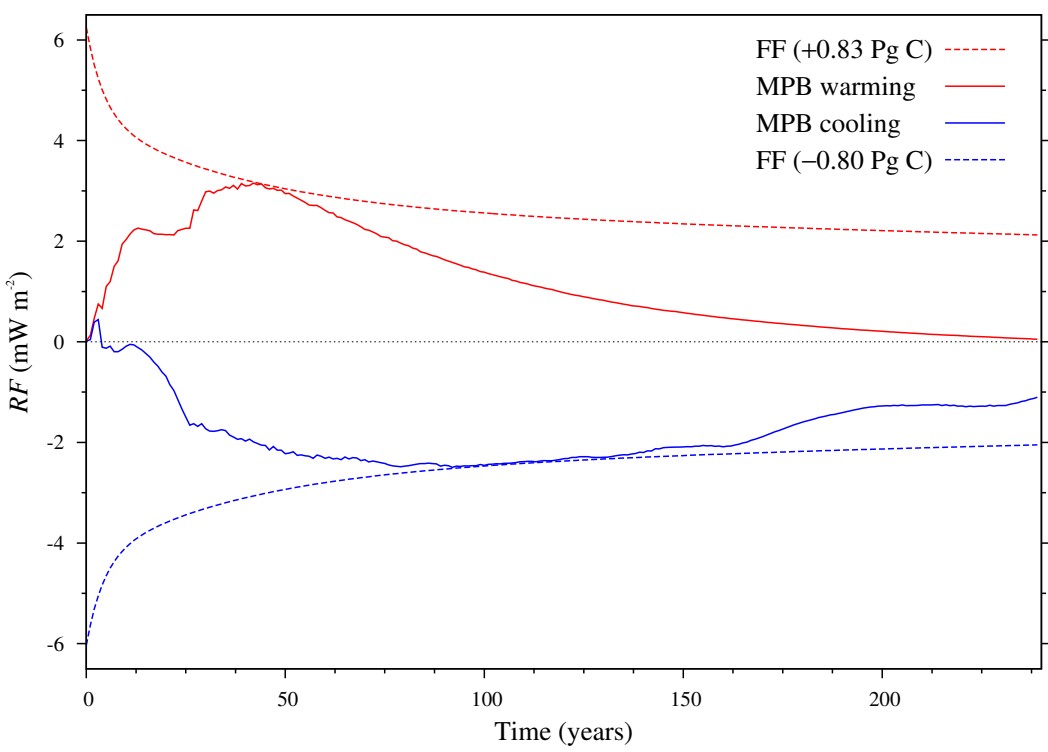

**Figure 7.** Comparison of the strongest warming and cooling radiative forcing ($RF$) responses from the MPB $Peak$ outbreaks with the $RF$ from a pulse of fossil fuel (FF) $CO_2$ emissions (in milli-W m$^{-2}$). The MPB $RF$ were computed for an outbreak area of 18.1 Mha; the warming response came from the NEonly scenario at the central location and the cooling response from the NE-LC scenario at the northern location.



**Table 1.** Input climate data and soil texture for the three locations.

| Element | Northern | Central | Southern |
|---|---|---|---|
| Temperature (°C) | | | |
| Annual | +0.7 | +0.8 | +2.5 |
| Dec–Feb | −11.3 | −8.8 | −6.8 |
| Mar–May | +0.9 | +0.4 | +2.0 |
| Jun–Aug | +11.9 | +9.9 | +12.0 |
| Sep–Nov | +1.0 | +1.4 | +2.7 |
| Precipitation (mm day$^{-1}$) | | | |
| Annual | 1.7 | 1.6 | 1.6 |
| Dec–Feb | 2.0 | 1.9 | 2.3 |
| Mar–May | 1.2 | 1.1 | 1.4 |
| Jun–Aug | 1.9 | 1.6 | 1.3 |
| Sep–Nov | 1.8 | 1.7 | 1.6 |
| Soil texture | Sandy loam | Loam | Sandy loam |
| Sand (%) | 65 | 42 | 65 |
| Silt (%) | 25 | 40 | 25 |
| Clay (%) | 10 | 18 | 10 |





**Table 2.** The four different scenarios simulated for the coexistence of plant functional types (PFTs). NE is needleleaf evergreen tree (i.e., the target PFT); LC is lower canopy (i.e., the sum of evergreen shrubs, cold-deciduous shrubs, and $C_3$ grasses); BD is broadleaf deciduous tree.

| Scenario | PFTs allowed |
|---|---|
| NEonly | NE |
| NE-LCcons | NE and LC, with constant LC biomass from the first outbreak onwards |
| NE-LC | NE and LC |
| AllPFT | NE, LC, and BD |