# Peer review of "Modelling long-term impacts of mountain pine beetle outbreaks on merchantable biomass, ecosystem carbon, albedo, and radiative forcing"

_Biogeosciences, 2016_

## Referee Comment (RC1) · Anonymous Referee #1 · 13 May 2016

General comments

This manuscript describes a modeling study on the impacts of mountain pine beetle (MPB), under various outbreak regimes and at three different sites, on biomass, carbon, and radiative forcing. This is an interesting topic and, as the authors note, one that has been little explored by high-caliber models such as the IBIS one used here. The ms is extremely well written, frequently insightful, and the technical approach used is solid.

There are a few weaknesses and unclear points. Most seriously, the authors need to

specify availability of the model code and outputs (I'd strongly suggest archiving the latter in a repository, and providing a link in the text). In addition, the sources and assumptions for the climate data used are not at all clear–were current-day conditions held constant? Was a specific climatology used? Finally, I'd suggest changing the conclusions and Figure 1 (see specific comments below).

In summary, this is a strong and well-done ms that needs only minor revisions for clarity and concision in a number of places.

Specific comments

1. Page 1, line 3: "non-target vegetation" – I understand, but slightly confusing. Reword if possible

2. P. 1, l. 11: contrary to what?

3. P. 3, l. 29-30: this is confusing, as "PFT" in this sentence is different from "PFT" in the previous sentence (I think). Clarify

4. P. 5, l. 19: model was spun up for a specific length of time, not based on stability of C pools? A few more details here would be useful

5. P. 5, l. 28-29: clarify a bit about where these values come from

6. P. 6: code availability of IBIS and MIM? What's availability of simulation results?

7. P. 9, l. 17-: well written and a good point. Maybe adapt as last sentence in abstract?

8. P. 12-13: I'm not a fan of restating the entire study in the conclusion. Consider Condensing the first two paragraphs to a couple sentences, and focusing on the third paragraph, which actually does contain conclusions

9. Figure 1 is not effective; I don't think readers need help locating Canada. Better would be to zoom in on BC, showing e.g. climate of the province with study locations marked

10. Table 1: where do these data come from?

---

## Referee Comment (RC2) · Anonymous Referee #2 · 20 May 2016

In most respects this is an excellent study demonstrating how the magnitude of carbon and albedo radiative forcings from MPB outbreaks depends on a range of site-specific factors, particularly the degree to which non-host trees and lower canopy trees support post-disturbance growth and carbon accumulation. This study has many strengths. The introductory framing is great, outlining the state of knowledge and setting up the present study. The modeling design is excellent, presenting a range of scenarios for post-disturbance forest growth and exploring also the effects of disturbance severity (intensity) and return interval. Radiative forcing (RF) results are presented in a very instructive and useful way, showing both time traces and also time-averages over the

240 year simulation period, and also providing results on a per hectare basis (with good discussion of scalability). Results on the radiative forcing from MPB outbreaks is compared to a fossil fuel (FF) pulse (or carbon sequestration event) by identifying the corresponding FF CO2 RF that would be of similar magnitude, providing a very thoughtful and helpful frame of reference. The narrative is fairly open and honest about assumptions and limitations of the modeling assumptions, explaining their likely implications for the study's results (but see recommendations below). The graphics are of high quality and easy to interpret and understand. The writing is clear. The discussion is comprehensive but succinct, including some indications of a management context for the paper's findings. It is rare to find papers that are so well laid out and well thought out, and studies as complete.

I have four main recommendations for how I think the work could be improved, hopefully lending credibility or at least helping readers to understand more of what underlies the results. They revolve around displaying, justifying and/or evaluating the model parameter or structural treatments or the associated output.

1) It would help if you would present the CO2-only, albedo-only, and combined components of RF for at least some of the cases if not all.

2) Among a number of important findings, this work suggests that MPB-induced albedo RF is much weaker than its CO2 RF, contrary to some past work, particularly O'Halloran et al.'s study. This seems to be attributed to perceived weaknesses in other studies. However, the present study does not provide a quantitative evaluation of its modeled post-MPB carbon stock changes or albedo changes, nor does it present the data and results that would be needed for others to be able to do so. Furthermore, it would be helpful to have a table or figure that provides quantitative comparisons to works by others. For example, how does this study's delta albedos in absolute units (not percentage changes) compare to those found by O'Halloran et al., Vanderhoof et al., Bright et al and/or other studies. Similarly for NEP, NPP, or carbon stock changes on a per area (e.g. g C m-2 y-1) basis compared to whatever is in the works of Hicke et

al., Romme et al., Kashian et al., Ghimire et al., Kurz et al. etc. This could be done with tables or figures but either way would provide an opportunity for readers to (a) see more of what is behind this study's results, and (b) have a better sense for how and why its findings differ from those reported previously. There are hints in the discussion but expansion in this way would help.

3) This studies most important finding is that carbon dynamics dominate the RF of MPB outbreaks in this region and that carbon dynamics can vary enormously from a new warming from a reduction in stocks to net cooling from an increase in carbon stocks if sequestration by surviving individuals outpaces that prior to the outbreak. This result hinges very importantly on the model's assumptions about the capacity for the lower canopy, for surviving trees, and for non-host PFTs to experience a release and experience vigorous regeneration and growth following MPB damage. Some of this modeled response is even stimulated by overestimation of heat storage in the model's treatment of dead standing trees (P11, L13), which seems odd. I am fine with the in-evitability of a model treatment of such dynamics that is a necessary simplification of reality. However, there is no presentation of the rate of post-disturbance growth for the various PFTs, and thus no way for readers to judge if the model's characterization of this post-D growth is plausible. The paper should show temporal trajectories of stand level biomass and NPP with time since disturbance for each PFT and for each case of the disturbance scenario by climate setting design. Ideally the authors would present data from forest inventory plots alongside this and any data on NPP with time since disturbance to provide a context for evaluation. It would also help if you would do more to explain and demonstrate what is behind the NPP increase that we see post-MPB oubreak for the LC unconstrained and AllPFT cases. Is there any observational or experimental evidence in this region and these forest types to support the model's out-come that stand level NPP is higher in stands that have a mix of PFTs than for NE-only stands, with all else being equal (climate, topographic setting, soil type and fertility)? It also seems like a significant weakness that the model's default PFT growth parameters were used for this study (P12, L16) without adjustment to be representative of productivity, allocation, turnover, and growth dynamics in this region's particular forest types. This should be avoided for regional applications, or at least the model's parameters and emergent growth dynamics should be evaluated against data.

4) Since this is a model, it should be possible to unambiguously explain precisely why the results turn out to be what they are, albeit with some additional simple point-scale model experiments. In a few cases key model outcomes remain incompletely understood and each should be documented more fully with further testing or demonstration. a) It is unclear exactly why the presence of dead standing trees leads to elevated productivity, or at least enhanced carbon uptake, for surviving trees either in the lower canopy or for non-host trees such as the broadleaf deciduous PFT. b) What explains the smaller post-disturbance productivity (and carbon stock) increase and release in the southern compared to central and northern settings remains unclear. The authors conjecture that this is due to soil moisture stress in the southern setting but precipitation is not much lower there and temperatures are not much hotter there (Table 1). Is there observational support for such a strong gradient in productivity and such strong soil moisture limitation in the southern region? c) It is unclear why climate effects are unimportant for the needleleaf evergreen PFT (only) but so pronounced when the lower canopy can respond in an unconstrained way and when all PFTs are present, particularly the broadleaf deciduous PFT. This should be explained with quantitative demonstration.

Other comments:

Application of the model's results with maps of MPB severity, pre-outbreak lodgepole and non-host density, and climate regimes would be a fantastic extension to translate the heuristic model based findings to an estimate of the landscape and region-wide carbon, albedo, and net RF implications of the outbreak. Not for this study of course.

The abstract should have some quantitative results (numbers) for example on the RF for a 1 ha outbreak of a certain kind (severity, model assumption) for CO2 only and

combined with albedo RF.

The paper has many abbreviations, some of which are less common. I recommend spelling it out in most cases, particularly for DST, maybe even NE and BD. It does not take much to do so and it is so much easier for readers.

P10, L5: It would help if you put the present result of 818 Tg C over 50 years into terms that are more comparable to those of the Kurz et al. 2008 study by making at least the time frame consistent (20 or 21 years only).

P13, L7: Conclusions: It would be helpful if you also noted that this is equivalent to about 4 years of Canada's emissions. The global context is a bit unfair as a way of judging the importance of a regional episode.

P4, L10+: Either here or in the discussion it might be helpful if you were to compare some of these model treatments/assumptions to what has been done in work by others who sought to model similar dynamics (e.g. Kurz et al., Ghimire et al, Arora et al, Edburg et al.).

---

## Referee Comment (RC3) · Anonymous Referee #3 · 27 May 2016

This study used a process-based model to address carbon and radiation balance of various beetle outbreak patterns and plant communities in stands historically dominated by lodgepole pine. They conclude that impact of MPB outbreak on carbon balance and radiative forcing varied depending on presence and response of non-target plants and report the resulting estimate of radiative forcing. I generally appreciated the clarity and contribution of the study, as I think some of the concepts and general demonstration of how such models can be used would be helpful to forest planners.

Main points:

[Figure]

My understanding is that the model represented competition during regrowth, but that establishment was prescribed. If I am incorrect about that, the authors could possibly try to mention that point early on and perhaps also outline why the model didn't represent establishment. I'm assuming lots of folks would be interested in establishment (in addition to the prescribed behaviours that were explored).

P7-L11: We believe? Wouldn't this be something worth confirming? What is the difference in air temperature, gas exchange, meristem activity, etc? Was there no other comparable literature on these microclimate effects? I seem to recall some good studies on microclimatic responses to clearcut vs. selective harvesting that might be worth comparing against.

Are the chosen prescribed plant community types representative of what is actually happening in response to the 1999- outbreak?

Should potential impacts on regional hydrology be factored into calculations of radiative forcing? Or does actual evapotranpiration and runoff remain stable through these outbreaks?

Were these experiments run under a historical level of $CO_2$ and N dep., or near future $CO_2$? Does that influence competition between the PFTs?

Technical:

P1-L11: on the contrary?

P1-L13-14: awkward wording

P12-L4: I was a bit confused by the statement because are other NE species not also being released? I thought there was a lot of subalpine fir coming up. I don't know that release is the contentious issue the authors make it out to be.

P14-L20: You could perhaps add a sentence explaining roughly what Kalb is representing.

Table 1: + symbols probably not necessary.

[Figure]

---

## Author Response (AR1)

July 18, 2016

Dr. Anja Rammig
Associate Editor
Biogeosciences

**RE: Submission of a revised manuscript to Biogeosciences**

Dear Dr. Rammig,

We are pleased to submit the revised version of our manuscript titled "Modelling long-term impacts of mountain pine beetle outbreaks on merchantable biomass, ecosystem carbon, albedo, and radiative forcing" (bg-2016-149) to Biogeosciences. As you could see in our previous point-by-point response, we made several major changes to address the comments from the three reviewers: we added a new "Comparison to previous studies" subsection, we produced a new Supplement with one figure and two tables, we included "Code availability" and "Data availability" sections, and we improved the text in all sections of the former manuscript.

Please find the following elements below:

- The reproduction of our previous point-by-point response to the reviewers' comments, as uploaded on the website of Biogeosciences on July 15th (including the new Supplement); and

- The 'track changes' version of our revised manuscript. There appears to be minor issues with tracking changes in the LaTeX package from Copernicus, so that the text is sometimes misaligned and a few additions (i.e., the "%" symbol, URLs, and new references in the bibliography) do not appear in blue. Finally, please note that we wrote "Tables S1-S2" in the revised manuscript when referring to the tables of the new Supplement, instead of "Tables S1-2" as in our previous response.

Best regards,

Jean-Sébastien Landry (on behalf of all co-authors)
McGill University (now at Concordia University)

July 15, 2016

Dr. Anja Rammig
Associate Editor
Biogeosciences

**RE: Response to comments from reviewers**

Dear Dr. Rammig,

We are pleased to submit our responses to the comments that the three reviewers provided on our manuscript titled "Modelling long-term impacts of mountain pine beetle outbreaks on merchantable biomass, ecosystem carbon, albedo, and radiative forcing".

All reviewers found the original manuscript technically solid and well written, and highlighted various contributions to the literature. One reviewer even stated that "[i]*t is rare to find papers that are so well laid out and well thought out, and studies as complete.*" You will find below how we further improved the manuscript based on their general comments and specific suggestions. You will also find at the end of our response the new Supplement (one figure and two tables) we produced to address some comments from Reviewer #2.

Finally, please note that all page and line numbers we provide below are for the original version of our manuscript.

Best regards,

Jean-Sébastien Landry (on behalf of all co-authors)
McGill University (now at Concordia University)

**Comments from Reviewer #1:** please note that the review of our manuscript is in italics, with our responses given in a regular font.

*General comments*

*This manuscript describes a modeling study on the impacts of mountain pine beetle (MPB), under various outbreak regimes and at three different sites, on biomass, carbon, and radiative forcing. This is an interesting topic and, as the authors note, one that has been little explored by high-caliber models such as the IBIS one used here. The ms is extremely well written, frequently insightful, and the technical approach used is solid.*

*There are a few weaknesses and unclear points. Most seriously, the authors need to specify availability of the model code and outputs (I'd strongly suggest archiving the latter in a repository, and providing a link in the text). In addition, the sources and assumptions for the climate data used are not at all clear–were current-day conditions held constant? Was a specific climatology used? Finally, I'd suggest changing the conclusions and Figure 1 (see specific comments below).*

*In summary, this is a strong and well-done ms that needs only minor revisions for clarity and concision in a number of places.*

>> 1.1 We thank you for this very positive assessment. You will see below how we responded to the few weaknesses and unclear points you identified to help us improve the manuscript. <<

*Specific comments*

*1. Page 1, line 3: "non-target vegetation" – I understand, but slightly confusing. Reword if possible*

>> 1.2 We agree that "non-target vegetation" (i.e., deciduous trees and lower-canopy shrubs and grasses, which are never targeted by MPB) is not a standard expression. However, we were unable to find a better one. "Surviving vegetation" could mislead readers into thinking that we also refer to target trees that survived a MPB attack, which is not the case. Similarly, "non-attacked vegetation" could mislead readers into thinking that we also refer to individual trees of the target species that were not attacked (e.g., because they are too small), which is not the case. Given that "non-target vegetation" is not standard, we made sure to define the expression upon its first use (in the Abstract) in our previous manuscript. To further prevent confusion, we added a similar clarification at the end of the Introduction (p3, l5; new text in blue): "Our purpose was not to forecast stand-level forest attributes (e.g., species-level basal area), but to contrast responses for very different scenarios about the presence and response of "non-target vegetation", which consisted of deciduous trees and lower-canopy shrubs and grasses that are never targeted by the MPB." <<

*2. P. 1, l. 11: contrary to what?*

>> 1.3 The purpose of "on the contrary" was to contrast the results for ecosystem carbon and radiative forcing with those, mentioned in the previous sentence, for merchantable biomass and surface albedo. We rephrased the text (p1, l8; modifications in blue and strikethrough) as: "The impacts of MPB outbreaks on merchantable biomass (decrease) and surface albedo (increase) were similar across the 12

combinations of locations and vegetation coexistence scenarios. The  impacts on ecosystem carbon and radiative forcing, however, varied substantially in magnitude and sign depending upon the presence and response of the non-target vegetation, particularly for the two locations not subjected to growing-season soil moisture stress; this variability represents the main finding from our study." <<

*3. P. 3, l. 29-30: this is confusing, as "PFT" in this sentence is different from "PFT" in the previous sentence (I think). Clarify*

>> 1.4 Throughout the text, "PFT" always means "plant functional type". We are unsure about the source of confusion, but we slightly rephrased the sentence to clarify our idea and address comment 3.2 from Reviewer #3 (p3, l28; modifications in blue and strikethrough): "For each PFT that can exist in the grid cell based on prevailing climatic conditions, leaf area index (LAI) cannot become lower than a very small, but non-zero, value; if a PFT undergoes 100% mortality in a grid cell (e.g, as occurred with our *Peak* regime; see Section 2.3), this "seed" LAI can therefore initiate regeneration. IBIS does not simulate establishment of many individuals for the same PFT in a grid cell." <<

*4. P. 5, l. 19: model was spun up for a specific length of time, not based on stability of C pools? A few more details here would be useful*

>> 1.5 Thank you for pointing this out. The 400-year spin up included an acceleration procedure for soil carbon pools and was therefore sufficient to reach stability. We added this clarification to the text (p5, l18; new text in blue): "For a given set, the five independent simulations branched from the same 400-year spin up (which was sufficient for carbon pools to stabilize; see Landry et al. (2016) for more details) and consisted of: [...]." <<

*5. P. 5, l. 28-29: clarify a bit about where these values come from*

>> 1.6 We introduced the following clarification (p5, l27; new text in blue): "Note that over these last 240 years, the mean mortality was 0.83% $yr^{-1}$ for the three periodic regimes (e.g., 16.6% mortality every 20 years for *Small*), thereby allowing the effect of outbreak severity vs. return interval to be compared for the same mean mortality, but was only 0.416% $yr^{-1}$ for the *Peak* regime (i.e., a single 100% mortality event over 240 years)." We mentioned in the Introduction (p3, l7) that these were "idealized outbreak regimes". <<

*6. P. 6: code availability of IBIS and MIM? What's availability of simulation results?*

>> 1.7 IBIS–MIM code is available online, the link being provided in the "Code availability" section of Landry et al. (2016). Here, we repeated this information in a new "Code availability" section after the Conclusions (p13, l18; new text in blue): "IBIS–MIM code is available upon request from the corresponding author or through the following link: http://landuse.geog.mcgill.ca/~jean-sebastien.landry2@mail.mcgill.ca/ibismim/." For the results, we added a new "Data availability" section after the new "Code availability" section: "Simulation results (as NetCDF files; http://www.unidata.ucar.edu/software/netcdf/) are available upon request from the corresponding

author." Please note there are 30 such files for each of the 60 simulations we performed (four vegetation coexistence scenarios, three locations, and five MPB outbreak regimes including the no-outbreak control). <<

*7. P. 9, l. 17-: well written and a good point. Maybe adapt as last sentence in abstract?*

>> 1.8 Thank you for this positive comment and the suggestion. We added the following sentence at the end of the Abstract: "Given that the variability of pre-outbreak vegetation characteristics can lead to very different regeneration pathways, the four vegetation coexistence scenarios we simulated probably only sampled the range of possible responses." <<

*8. P. 12-13: I'm not a fan of restating the entire study in the conclusion. Consider Condensing the first two paragraphs to a couple sentences, and focusing on the third paragraph, which actually does contain conclusions*

>> 1.9 We believe that some readers might initially only look at the Abstract, Introduction, and Conclusions to decide if reading the entire paper is worthwhile. Therefore, we think that highlighting the main outcomes, along with the corresponding figures, can be helpful. Nonetheless, we did substantially shorten the text as suggested by combining the first two paragraphs into a single one (p12, l23; modified text in blue and strikethrough): "Despite major progress over the last decades, various knowledge gaps still limit the understanding of the consequences of mountain pine beetle (MPB) outbreaks. In this study, we used a climate-driven process-based ecosystem model to estimate the long-term impacts of prescribed MPB outbreaks ~~on four variables relevant to the forestry sector, land-atmosphere exchanges of carbon and energy, and global climate change, while explicitly accounting for: (1) different vegetation coexistence scenarios and strengths of the post-outbreak growth release, including lower-canopy shrubs and grasses (Table 2); (2) different outbreak severities and return intervals; (3) the long-term effect of repeated outbreaks; (4) the biophysical influence of MPB-killed dead standing trees (DSTs); and (5) the effect of changes in surface albedo (α) on the net radiative forcing (RF) resulting from MPB outbreaks. Using a climate-driven process-based model further allowed us to compare responses across three locations in British Columbia (Fig. 1) having different climatic conditions (Table 1)(following MPB mortalitymuchWe also found that higher outbreak severity (i.e., per-event mortality) usually caused a stronger mean long-term effect than a shorter return interval between outbreaks (Fig. 6) and that, by warming the canopy as a result of radiation absorption, DSTs might increase the productivity of surviving and regrowing trees.~~" In the last paragraph of the Conclusions (p13, l13), we therefore replaced the "DSTs" abbreviation by "dead standing trees". <<

*9. Figure 1 is not effective; I don't think readers need help locating Canada. Better would be to zoom in on BC, showing e.g. climate of the province with study locations marked*

>> 1.10 The purpose of Figure 1 is to help readers locate British Columbia within Canada, which might be particularly useful for readers outside North America. Moreover, Figure 1 helps readers appreciate the size of British Columbia vs. Canada or the United States. When seeing Figure 1 after reading that "in British Columbia [...] more than half of the merchantable pine volume has been killed" by MPB, readers should thus better grasp the magnitude of the outbreak. About climate, we consider that Table 1 is more efficient than a figure to provide all the relevant data (annual and seasonal averages for both temperature and precipitation) for the three locations studied. The reason to add a climate map of British Columbia would seem to be for the purpose of verifying if our three locations are representative of the province's climate, which is not what we aimed for. As mentioned in the manuscript (p4, l33), we rather chose these locations for the following reasons: 1) having experienced noticeable MBP mortality since 2000; and 2) having different climates. <<

*10. Table 1: where do these data come from?*

>> 1.11 We did not previously provide the references. Here is how we addressed this weakness (p4, l29; new text in blue): "For all variables, including the ones provided in Table 1, we used the same input data as Landry et al. (2016): the mean 1961–1990 atmospheric $CO_2$ level, gridded 1961–1990 monthly mean data for climate (New et al., 1999), and survey data from versions 2.1 and 2.2 of the Soil Landscapes of Canada for soil (http://sis.agr.gc.ca/cansis/nsdb/slc/index.html)." <<

**References**

Landry et al. (2016). Geoscientific Model Development 9, 1243-1261
New et al. (1999). Journal of Climate 12, 829-856

**Comments from Reviewer #2:** please note that the review of our manuscript is in italics, with our responses given in a regular font.

*In most respects this is an excellent study demonstrating how the magnitude of carbon and albedo radiative forcings from MPB outbreaks depends on a range of site-specific factors, particularly the degree to which non-host trees and lower canopy trees support post-disturbance growth and carbon accumulation. This study has many strengths. The introductory framing is great, outlining the state of knowledge and setting up the present study. The modeling design is excellent, presenting a range of scenarios for post-disturbance forest growth and exploring also the effects of disturbance severity (intensity) and return interval. Radiative forcing (RF) results are presented in a very instructive and useful way, showing both time traces and also time-averages over the 240 year simulation period, and also providing results on a per hectare basis (with good discussion of scalability). Results on the radiative forcing from MPB outbreaks is compared to a fossil fuel (FF) pulse (or carbon sequestration event) by identifying the corresponding FF CO2 RF that would be of similar magnitude, providing a very thoughtful and helpful frame of reference. The narrative is fairly open and honest about assumptions and limitations of the modeling assumptions, explaining their likely implications for the study's results (but see recommendations below). The graphics are of high quality and easy to interpret and understand. The writing is clear. The discussion is comprehensive but succinct, including some indications of a management context for the paper's findings. It is rare to find papers that are so well laid out and well thought out, and studies as complete.*

*I have four main recommendations for how I think the work could be improved, hopefully lending credibility or at least helping readers to understand more of what underlies the results. They revolve around displaying, justifying and/or evaluating the model parameter or structural treatments or the associated output.*

>> 2.1 We thank you for this very supportive review of our study. Please find below how we responded to the four main recommendations and other comments you provided to improve the manuscript. <<

*1) It would help if you would present the CO2-only, albedo-only, and combined components of RF for at least some of the cases if not all.*

>> 2.2 Thank you for this suggestion. We added a new Supplement to the manuscript, with a new figure showing the three *RF* components ($CO_2$, albedo, and total) for the *Peak* regime (i.e., the single 100% mortality event). We referred to this new figure on three instances. First, in section 3.1 on transient results (p7, l32; new text in blue): "*RF* varied substantially across locations and vegetation coexistence scenarios (Fig. 5; also see Fig. S1 in the Supplement for the $CO_2$-based and α-based components of the total *RF* response for *Peak*). Second, in section 3.2 on mean effects (p8, l17; new text in blue): "Figure 6 also suggests that *RF* was more closely linked to $\Delta C_{eco}$ than to $\Delta\alpha$, which is supported by the transient results where *RF* basically mirrored $\Delta C_{eco}$ (compare Figs. 3 and 5; also see Fig. S1)." Please see our response 2.3 below for the third mention of Fig. S1. <<

*2) Among a number of important findings, this work suggests that MPB-induced albedo RF is much weaker than its CO2 RF, contrary to some past work, particularly O'Halloran et al.'s study. This seems to be attributed to perceived weaknesses in other studies. However, the present study does not provide a quantitative evaluation of its modeled post-MPB carbon stock changes or albedo changes, nor does it*

*present the data and results that would be needed for others to be able to do so. Furthermore, it would be helpful to have a table or figure that provides quantitative comparisons to works by others. For example, how does this study's delta albedos in absolute units (not percentage changes) compare to those found by O'Halloran et al., Vanderhoof et al., Bright et al and/or other studies. Similarly for NEP, NPP, or carbon stock changes on a per area (e.g. g C m-2 y-1) basis compared to whatever is in the works of Hicke et al., Romme et al., Kashian et al., Ghimire et al., Kurz et al. etc. This could be done with tables or figures but either way would provide an opportunity for readers to (a) see more of what is behind this study's results, and (b) have a better sense for how and why its findings differ from those reported previously. There are hints in the discussion but expansion in this way would help.*

>> 2.3 We thank you for this suggestion, which stimulated us to further compare our results with previous studies in two Tables we added to the new Supplement (we address below the comment related to O'Halloran et al., 2012). We restricted these comparisons to empirical studies including no-outbreak controls, except for the modelling study of Arora et al. (2016) because it used a model (CLASS-CTEM) similar to IBIS-MIM and provided results for one cell located close to our central location. At the onset, we stress that these comparisons can only be approximate due to major differences in location (most studies were performed in the United States), cumulative mortality level (which often fell between our *Large* and *Peak* simulations), and temporal pattern of mortality (for simplicity we simulated one-year events because we were interested in long-term results, but actual MPB outbreaks in a given stand usually occur over many years). Furthermore, empirical studies on net MPB effects are themselves uncertain (e.g., finding appropriate control stands is challenging) as shown by very large error bars (e.g., Table 3 of Morehouse et al. (2008) for a field study and Fig. 2 of Bright et al. (2013) for a satellite study). With these caveats in mind, IBIS-MIM results agreed reasonably well with previous studies for different carbon cycle variables and for the change in albedo.

For carbon (Table S1), we performed the comparison based on the range of the 12 combinations of locations and vegetation coexistence scenarios for the outbreak regime (*Peak*, *Large*, or *Medium*) most similar to the cumulative mortality of each previous study. For the Arora et al. (2016) modelling study, we also showed IBIS-MIM results at the central location for the NEonly coexistence scenario (similar to the modelling approach they used). For all comparisons, results from IBIS-MIM vs. previous empirical studies overlapped and had a similar magnitude, with two exceptions. First, the mean change in $C_{eco}$ from Morehouse et al. (2008) was an order of magnitude larger than the mean result from IBIS-MIM. However, the very large value they obtained (-1.9 kgC/m2) 1-3 years only after mortality does not seem realistic and likely reflects the major uncertainties involved, as indicated by their large error bars and the fact that the change in $C_{eco}$ between stands with and without MPB attack was not "statistically significant" despite such a magnitude. Second, IBIS-MIM reduction in GPP 1-3 years after MPB mortality was noticeably stronger than the results from Bright et al. (2013). However this last study defined the start of MPB attack as the year when mortality was >1%, hence their mean results included many stands barely impacted yet. For years 4-9, when actual mortality was much higher, mean results from IBIS-MIM and Bright et al. (2013) were almost equal. Similarly, differences in the temporal pattern of mortality likely explain why IBIS-MIM results were overall similar to the ones from Arora et al. (2016)—especially for NEonly at the central location—but with a faster 'decrease-and-recovery' pattern. Indeed, the ~80% mortality occurred over ~10 years in Arora et al. (2016) (see their Fig. 1b-e), compared to the single-year 100% mortality we simulated under the *Peak* regime. For the change in albedo (Table S2), results from IBIS-MIM vs. previous empirical studies also generally overlapped and had a similar magnitude. The few discrepancies once again likely involve a different temporal pattern of mortality as well as a quicker reestablishment of forest cover in the United States vs. British Columbia. (For results that were mean values over many months, please note that we simply averaged the different monthly values (i.e., we did not weight them by the incoming solar

radiation, as should normally be done when reporting mean albedo values over many months); previous studies did not report how they computed their mean values. For Vanderhoof et al. (2014), we performed the comparisons based on MODIS results even though they have a coarser spatial resolution than Landsat results, because the latter were considered "experimental" in the presence of snow.)

We modified the text in two places to introduce this new analysis (modified text in blue and strikethrough). First, in Section 2.2 (p4, l16): "IBIS–MIM results after a simulated MPB outbreak generally agreed with previous  studies  (Landry et al., 2016)." Second, we added a new subsection "3.1 Comparison to previous studies" before showing the results (p7, l2): "IBIS–MIM results following a MPB outbreak have already been found to agree with results from 38 field-, satellite-, and model-based studies for 28 variables related to vegetation dynamics and the exchanges of carbon, energy, and water, over daily to multi-annual timescales (Landry et al., 2016). That previous assessment also provided time-since-disturbance NPP results for different PFTs under the NE-LCcons scenario. Here we performed an additional assessment of IBIS–MIM, comparing its results for different variables related to the carbon cycle and for $\Delta\alpha$ to previous studies (see Tables S1-2 in the Supplement). We restricted these comparisons to empirical studies that included control stands, except for the modelling study of Arora et al. (2016) that used a model similar to IBIS–MIM under a NEonly-type setting and provided results for one grid cell close to our central location. Given the differences in locations, cumulative mortality levels, and temporal patterns of mortality, IBIS–MIM agreed reasonably well with previous studies, providing a measure of confidence in the realism of the results shown below."

About our finding that MPB-caused albedo *RF* was weaker than the associated $CO_2$ *RF*, we want to clarify two points. First, to our knowledge O'Halloran et al. (2012) is the only previous study that addressed this question. Other studies comparing albedo *RF* to $CO_2$ *RF* were for changes other than MPB outbreaks: permanent changes in land cover (Betts, 2000; Lohila et al., 2010; Bernier et al., 2011; Pongratz et al., 2011) or transient changes caused by fire, hurricane, or forest biofuels (Randerson et al., 2006; Bright et al., 2011; O'Halloran et al., 2012; Landry et al., 2016b). Therefore, our results are only comparable to those of O'Halloran et al. (2012) for MPB. Second, the issue we identified with O'Halloran et al. (2012) is not merely a "*perceived weakness*[...]", but a major spatiotemporal mismatch in their albedo *RF* vs. $CO_2$ *RF* comparison. Indeed, their albedo *RF* came from stand-level time-since-disturbance results restricted to stands that underwent substantial mortality, but their $CO_2$ *RF* was based on mean regional-level results from a modelling study over 374,000 km2 in which <30% of the total area (cumulative value) had been attacked by Year 5, with most of this area undergoing "Light" and "Moderate" mortality levels for which the fraction of softwood biomass actually killed was only 5% and 10%, respectively (see Fig. 3b and Supplementary Information of Kurz et al., 2008). We therefore challenge the conclusions of O'Halloran et al. (2012) about albedo *RF* always being stronger than $CO_2$ *RF* for the first 14 years post-mortality. However, we do not wish to convey either that $CO_2$ *RF* is necessarily always stronger than albedo *RF* following MPB outbreaks; in fact, our results sometimes show the opposite. We therefore modified the text as follows (p9, l30; new text in blue): "This outcome is at odds with IBIS–MIM results, because our net *RF* depended critically upon the sign of the $C_{eco}$ change and, even for instances of overall net cooling, our *RF* values were positive during the first four years at least (see transient results in panels g, h, j, and k of Fig. S1 for instances of $\alpha$-based cooling being temporarily stronger than $C_{eco}$-based warming under *Peak*)." <<

*3) This studies most important finding is that carbon dynamics dominate the RF of MPB outbreaks in this region and that carbon dynamics can vary enormously from a new warming from a reduction in*

*stocks to net cooling from an increase in carbon stocks if sequestration by surviving individuals outpaces that prior to the outbreak. This result hinges very importantly on the model's assumptions about the capacity for the lower canopy, for surviving trees, and for non-host PFTs to experience a release and experience vigorous regeneration and growth following MPB damage. Some of this modeled response is even stimulated by overestimation of heat storage in the model's treatment of dead standing trees (P11, L13), which seems odd. I am fine with the inevitability of a model treatment of such dynamics that is a necessary simplification of reality. However, there is no presentation of the rate of post-disturbance growth for the various PFTs, and thus no way for readers to judge if the model's characterization of this post-D growth is plausible. The paper should show temporal trajectories of stand level biomass and NPP with time since disturbance for each PFT and for each case of the disturbance scenario by climate setting design. Ideally the authors would present data from forest inventory plots alongside this and any data on NPP with time since disturbance to provide a context for evaluation. It would also help if you would do more to explain and demonstrate what is behind the NPP increase that we see post-MPB oubreak for the LC unconstrained and AllPFT cases. Is there any observational or experimental evidence in this region and these forest types to support the model's outcome that stand level NPP is higher in stands that have a mix of PFTs than for NE-only stands, with all else being equal (climate, topographic setting, soil type and fertility)? It also seems like a significant weakness that the model's default PFT growth parameters were used for this study (P12, L16) without adjustment to be representative of productivity, allocation, turnover, and growth dynamics in this region's particular forest types. This should be avoided for regional applications, or at least the model's parameters and emergent growth dynamics should be evaluated against data.*

>> 2.4 We respond to the many comments made above in the order they appear.

a) The variation in carbon dynamics did not depend much on the response of the "*surviving trees*" (changes in $C_{eco}$ were always negative for NEonly, which simulated the attacked PFT only; see Fig. 3a-c), but mostly on the response of non-target PFTs, including the lower-canopy grasses and shrubs.

b) The warming effect of the physical presence of dead standing trees, which do absorb incoming radiation, seems to substantially affect the recovery of targeted NE trees only (p7, l8). Consequently, it likely had a marginal effect on $C_{eco}$ (see "a)" above). The fact that IBIS overestimates heat storage in tree stems does not necessarily imply that this effect is a modelling artifact, but it does call for validation through field studies as we suggested (p11, l15).

c) Readers can look at Landry et al. (2016) for time-since-disturbance NPP results for different PFTs; we added this clarification to the new subsection 3.1 (see our response 2.3 above). Although lacking no-outbreak controls, three studies using the eddy covariance method support the realism of our NEP (which is more relevant than NPP for assessing changes in $C_{eco}$) results. Brown et al. (2012) found that annual NEP was positive or close to zero 3-5 years only after the onset of MPB mortality, for two stands in British Columbia <100 km from our northern location; they attributed this carbon response to surviving trees and vegetation. Studying the same two stands, Bowler et al. (2012) concluded that non-tree species were responsible for one to two thirds of post-MPB net photosynthesis. Reed et al. (2014) found that growing-season NEP remained stable over a 3-year period for a stand in Wyoming (United States) despite concurrent increases in MPB mortality, from 38% to 68%. All these studies support the realism of IBIS-MIM growth release and are often mentioned in our manuscript (p2, l11; p2, l24; p9, l14-15; p10, l34; p11, l1; and p11, l30).

d) About providing additional comparisons with other studies or data, the purpose of this study was not to perform such in-depth analyses. IBIS itself has already been compared to "empirical data over large

regions and several field sites, including in Canada" as shown in the seven studies mentioned in the text (p3, l31). Landry et al. (2016) presented a detailed assessment of IBIS-MIM, and here we added Tables S1-2 to compare IBIS-MIM results for carbon and albedo to seven other studies—covering all the empirical studies of which we are aware and that directly quantified MPB effects by comparison with no-outbreak controls. Also note that previous modelling studies quantifying MPB impacts on $C_{eco}$ or NEP (Kurz et al., 2008; Edburg et al., 2011; Ghimire et al., 2015; Arora et al., 2016) did not present such detailed comparisons with empirical data on MPB effects.

e) We clearly acknowledged that "IBIS parameters for the different PFTs were also not specifically based on data gathered from British Columbian forests" (p12, l16). Despite this limitation, our NEonly results appear consistent with the ones from Arora et al. (2016) (see Table S1) who used the "Interior needleleaf evergreen" PFT developed for interior British Columbia by Peng et al. (2014). In fact, our results suggest that accounting for different vegetation coexistence scenarios has a much greater impact on simulated post-MPB $C_{eco}$ in models like IBIS-MIM and CLASS-CTEM than using different (but reasonable) values for PFT parameters. Note that for the MPB-attacked PFT, >50 parameters influenced the exchanges carbon, energy, water, and momentum in IBIS-MIM. <<

*4) Since this is a model, it should be possible to unambiguously explain precisely why the results turn out to be what they are, albeit with some additional simple point-scale model experiments. In a few cases key model outcomes remain incompletely understood and each should be documented more fully with further testing or demonstration. a) It is unclear exactly why the presence of dead standing trees leads to elevated productivity, or at least enhanced carbon uptake, for surviving trees either in the lower canopy or for non-host trees such as the broadleaf deciduous PFT. b) What explains the smaller post-disturbance productivity (and carbon stock) increase and release in the southern compared to central and northern settings remains unclear. The authors conjecture that this is due to soil moisture stress in the southern setting but precipitation is not much lower there and temperatures are not much hotter there (Table 1). Is there observational support for such a strong gradient in productivity and such strong soil moisture limitation in the southern region? c) It is unclear why climate effects are unimportant for the needleleaf evergreen PFT (only) but so pronounced when the lower canopy can respond in an unconstrained way and when all PFTs are present, particularly the broadleaf deciduous PFT. This should be explained with quantitative demonstration.*

>> 2.5 We respond to the many comments made above using the same letters.

a) There seems to be some confusion about what we reported. We did not report that the warming effect from the physical presence of dead standing trees increased productivity of the lower canopy or broadleaf deciduous trees, but only for "regrowing NE trees" (p7, l8). Please remember that "lower canopy" designates shrubs and grasses only (as mentioned on p1, l3; p2, l25; p3, l22; p5, l6; and in Table 2).

b) Note that we did not actually claim there was "*a strong gradient in productivity*" under normal circumstances, but simply that the post-MPB growth release was weaker at the southern location. This is indeed likely explained by growing-season soil moisture stress, because simulated "plant available water" (which accounts for water distribution throughout the different soil layers) during the growing season at the southern location was often low enough to reduce productivity (results not shown). At the northern location, summer precipitation is 46% higher than at the southern location for a similar temperature; at the central location, summer precipitation is 23% higher, combined with much lower temperature and a less sandy soil (Table 1). Climate is notoriously drier around the semi-arid southern

location (close to Kelowna) than over most of British Columbia; this is actually the reason why we chose this specific location to assess the influence of climatic conditions.

c) We agree that it would be more satisfying to investigate the exact reasons why results were so similar across locations for the NEonly vegetation scenario. However, we unfortunately cannot pinpoint these reasons without doing lots of additional simulations going beyond the scope of this study. This would end up being very challenging given that exchanges of carbon, energy, and water are all 'tied together' in IBIS... not to mention that all these exchanges were computed with a 60-minute time step during 640 years (400 years spin-up plus 240 years after the first MPB outbreak)! <<

*Other comments:*

*Application of the model's results with maps of MPB severity, pre-outbreak lodgepole and non-host density, and climate regimes would be a fantastic extension to translate the heuristic model based findings to an estimate of the landscape and region-wide carbon, albedo, and net RF implications of the outbreak. Not for this study of course.*

>> 2.6 We thank you for this suggestion. <<

*The abstract should have some quantitative results (numbers) for example on the RF for a 1 ha outbreak of a certain kind (severity, model assumption) for CO2 only and combined with albedo RF.*

>> 2.7 The fact that we performed 48 different MPB outbreak simulations (three locations, four vegetation coexistence scenarios, and four outbreak regimes) over 240 years for four different variables makes this complicated: providing quantitative results with the required context would take too much space and/or involve arbitrary choices. Please remember that "[w]e do not believe that any of the vegetation coexistence scenarios we simulated is fundamentally more realistic than the others" (p.9, l17) and that results sometimes changed sign depending upon the vegetation coexistence scenario. <<

*The paper has many abbreviations, some of which are less common. I recommend spelling it out in most cases, particularly for DST, maybe even NE and BD. It does not take much to do so and it is so much easier for readers.*

>> 2.8 We agree that fewer abbreviations might increase readability, but we decided to not make these changes. Please note that the NE, LC, BD, and related scenario abbreviations are already defined in Table 2, while many other abbreviations (MPB, PFT, LAI, NPP, NEP, $CO_2$, and GHG) are standard in the field. Other abbreviations (IBIS, MIM, DST, IFT, $C_{eco}$, $B_{merch}$, $\alpha$, and $RF$) are less common, but always spelling out all abbreviations (vs. only the first time they appear and in the Conclusions, as we did) would substantially lengthen the manuscript. <<

*P10, L5: It would help if you put the present result of 818 Tg C over 50 years into terms that are more comparable to those of the Kurz et al. 2008 study by making at least the time frame consistent (20 or 21 years only).*

>> 2.9 We thank you for this suggestion. The text now reads (p10, l4; new text in blue): "For the warming case, the maximum decrease in $C_{eco}$ (based on *Peak* from Fig. 3b, over 18.1 Mha) was equal to 818 Tg C ~50 years after mortality; for this same case, the decrease 21 years after mortality was 490 Tg C." <<

*P13, L7: Conclusions: It would be helpful if you also noted that this is equivalent to about 4 years of Canada's emissions. The global context is a bit unfair as a way of judging the importance of a regional episode.*

>> 2.10 We agree that the "*global context is a bit unfair*", yet we prefer not to add the Canada-only comparison for two reasons. First, our *RF* results were appropriate to bound the maximum value of the net warming or cooling impact caused by MPB, but not to assess its precise value. A multi-century effect that is less than one month of global anthropogenic $CO_2$ emissions is clearly 'small', no matter if the correct value is actually one day or three weeks. But for Canada only, the uncertainty associated with our bounding analysis is important: less than four years could be highly relevant (e.g., if the correct value is three years) or not (e.g., if the correct value is three months). Second, although "*unfair*" the global context is relevant because, in the wake of Kurz et al. (2008) who concluded that the current MPB outbreak was (emphasis is ours) "one example of how climate change can affect disturbance regimes for which impacts on the global carbon cycle then provide **strong positive feedback to the global climate system**", many scientific studies assume or state that MPB has a large impact on the global climate. <<

*P4, L10+: Either here or in the discussion it might be helpful if you were to compare some of these model treatments/assumptions to what has been done in work by others who sought to model similar dynamics (e.g. Kurz et al., Ghimire et al, Arora et al, Edburg et al.).*

>> 2.11 Thank you for this helpful suggestion. We added the following sentence after explaining our four vegetation coexistence scenarios (p5, l16): "Previous modelling studies quantifying MPB impacts on $C_{eco}$ or NEP (Kurz et al., 2008; Edburg et al., 2011; Ghimire et al., 2015; Arora et al., 2016) have resorted to NEonly-type approaches that did not account for the possible growth release of the non-target vegetation". <<

**Comments from Reviewer #3:** please note that the review of our manuscript is in italics, with our responses given in a regular font.

*This study used a process-based model to address carbon and radiation balance of various beetle outbreak patterns and plant communities in stands historically dominated by lodgepole pine. They conclude that impact of MPB outbreak on carbon balance and radiative forcing varied depending on presence and response of non-target plants and report the resulting estimate of radiative forcing. I generally appreciated the clarity and contribution of the study, as I think some of the concepts and general demonstration of how such models can be used would be helpful to forest planners.*

>> 3.1 We thank you for your appreciation of our study. You will find below our responses to your comments. <<

*Main points:*

*My understanding is that the model represented competition during regrowth, but that establishment was prescribed. If I am incorrect about that, the authors could possibly try to mention that point early on and perhaps also outline why the model didn't represent establishment. I'm assuming lots of folks would be interested in establishment (in addition to the prescribed behaviours that were explored).*

>> 3.2 Please note this mainly arises from the class of models to which IBIS belongs: IBIS is not a gap model (as noted on p4, l29) and does not really represent establishment because it does not simulate multiple individual plants for the same PFT (as noted on p12, l12). When starting from bare ground as for the 400-year spin up, or when a PFT undergoes 100% mortality as simulated under the *Peak* regime, the PFT is given a very small 'seed' leaf area index to be able to perform photosynthesis and grow. We clarified these ideas, also addressing comment 1.4 from Reviewer #1 (p3, l28; modifications in blue and strikethrough): "For each PFT that can exist in the grid cell based on prevailing climatic conditions, leaf area index (LAI) cannot become lower than a very small, but non-zero, value; if a PFT undergoes 100% mortality in a grid cell (e.g, as occurred with our *Peak* regime; see Section 2.3), this "seed" LAI can therefore initiate regeneration. IBIS does not simulate establishment of many individuals for the same PFT in a grid cell." <<

*P7-L11: We believe? Wouldn't this be something worth confirming? What is the difference in air temperature, gas exchange, meristem activity, etc? Was there no other comparable literature on these microclimate effects? I seem to recall some good studies on microclimatic responses to clearcut vs. selective harvesting that might be worth comparing against.*

>> 3.3 We would like to clarify that this part of the text deals with a feature we observed in the modelling results from IBIS-MIM, not in empirical results. Therefore, other studies cannot tell us what is the source of this feature: for this, we can only look at IBIS-MIM results and how the model works. We have good reasons to believe that the dip in the recovery of NE trees for *Peak* was related to the loss of the warming effect when dead standing trees (DSTs), which do absorb incoming radiation in IBIS-MIM, were simulated to fall. We had a look at temperature results, which are consistent with this hypothesis. (Looking at simulated $CO_2$ exchanges would not be helpful, because these exchanges are the results we are trying to explain; as for meristem activity, it is not simulated by IBIS.) However, confirming this hypothesis would require us to perform other simulations in which everything would be

the same (e.g., DSTs would still shadow the regrowing NE trees and would still intercept precipitation), except that DSTs would not affect air temperature. Given the hundreds of coupled equations that simulate exchanges of carbon, energy, and water in IBIS-MIM, performing such simulations would represent a major coding and analysis challenge that is outside the scope of the current study. We would like to stress that IBIS is a computationally expensive ecosystem model, unlike some other models that are commonly used to study forests. <<

*Are the chosen prescribed plant community types representative of what is actually happening in response to the 1999- outbreak?*

>> 3.4 Although most studies focussed on what happens after MPB outbreaks, there is evidence that pre-outbreak plant communities in British Columbia are highly variable in terms of composition, sometimes due to the impacts of previous MPB outbreaks (Axelson et al., 2009; Amoroso et al., 2013; Alfaro et al., 2015; Campbell and Antos, 2015). Please note that we did not aim to tweak the model to obtain specific mixes of PFTs before the outbreaks, but rather let the different PFTs compete as simulated by IBIS for each vegetation coexistence scenario. These scenarios were very different in terms of presence and response of the non-target vegetation (e.g., from only NE trees to a mix of NE trees, BD trees, and various lower-canopy PFTs). <<

*Should potential impacts on regional hydrology be factored into calculations of radiative forcing? Or does actual evapotranpiration and runoff remain stable through these outbreaks?*

>> 3.5 This is a somewhat tricky question. MPB outbreaks in IBIS-MIM led to complex changes in the water cycle, as detailed in Landry et al. (2016). Changes in evapotranspiration are not traditionally included in the computation of radiative forcing, the reason being they do not directly alter Earth's boundary conditions but only represent a transfer of energy within the Earth system (the heat required for evapotranspiration at one location is released elsewhere upon precipitation). However, this traditional view overlooks two elements. First, changes in evapotranspiration can modify the average amount of atmospheric water vapour, which is a greenhouse gas. Second, and more importantly, changes in evapotranspiration can modify cloud cover, thereby affecting the Earth's energy budget (Ban-Weiss et al., 2011). However, these effects appear relatively small and, especially for the second effect, highly uncertain. For example, a study considering changes in evapotranspiration from basically all biomass burning and human activities except irrigation concluded that the resulting global cooling was not statistically significant (Jacobson, 2014). Irrigation might be the only change in evapotranspiration that leads to a global climate impact that is not negligible and that can be evaluated with reasonable confidence (Puma and Cook, 2010), but its radiative forcing is not quantified in IPCC reports (Myhre et al., 2013). To address this point, we added the following sentence when discussing the study limitations (p12, l21): "MPB outbreaks can also affect the global climate through other mechanisms that are poorly constrained and that we did not consider, for example changes in atmospheric chemistry (Arneth and Niinemets, 2010) or in the partitioning between latent and sensible heat fluxes (Ban-Weiss et al., 2011)". <<

*Were these experiments run under a historical level of CO2 and N dep., or near future CO2? Does that influence competition between the PFTs?*

>> 3.6 Thank you for this question. Please note that IBIS does not account for nitrogen (p12, l7). About $CO_2$ levels, we clarified the text (p4, l29; new text in blue): "For all variables, including the ones provided in Table 1, we used the same input data as Landry et al. (2016): the mean 1961–1990 atmospheric $CO_2$ level, gridded 1961–1990 monthly mean data for climate (New et al., 1999), and survey data from versions 2.1 and 2.2 of the Soil Landscapes of Canada for soil (http://sis.agr.gc.ca/cansis/nsdb/slc/index.html)." Using projected $CO_2$ levels would probably have hastened the recovery of NE trees and could have influenced competition among PFTs, but we do not think that this would fundamentally change our main finding about the role of the growth release from the non-target vegetation in modulating MPB effects on ecosystem carbon and radiative forcing. Note that $CO_2$ fertilization used to be notoriously stronger in IBIS than in other similar models (McGuire et al., 2001; Friedlingstein et al., 2006), but the improved leaf-to-canopy photosynthesis scaling procedure we introduced (Landry et al., 2016) reduced $CO_2$ fertilization strength in the model. <<

*Technical:*

*P1-L11: on the contrary?*

>> 3.7 The purpose of "on the contrary" was to contrast the results for ecosystem carbon and radiative forcing with those, mentioned in the previous sentence, for merchantable biomass and surface albedo. We rephrased the text (p1, l8; modifications in blue and strikethrough) as: "The impacts of MPB outbreaks on merchantable biomass (decrease) and surface albedo (increase) were similar across the 12 combinations of locations and vegetation coexistence scenarios. The  impacts on ecosystem carbon and radiative forcing, however, varied substantially in magnitude and sign depending upon the presence and response of the non-target vegetation, particularly for the two locations not subjected to growing-season soil moisture stress; this variability represents the main finding from our study." <<

*P1-L13-14: awkward wording*

>> 3.8 We rephrased the sentence (p1, l12; modifications in blue and strikethrough) as: "Despite major uncertainty in the value of the resulting radiative forcing, a simple analysis also suggested that the MPB outbreak in British Columbia will have a smaller impact on global temperature over the coming decades and centuries than a single month of global anthropogenic $CO_2$ emissions from fossil fuel combustion and cement production." <<

*P12-L4: I was a bit confused by the statement because are other NE species not also being released? I thought there was a lot of subalpine fir coming up. I don't know that release is the contentious issue the authors make it out to be.*

>> 3.9 Please remember that IBIS simulates a single NE PFT (p5, l5), which broadly represents many 'functionally equivalent' species in terms of land-atmosphere exchanges; since the NE PFT corresponded to MPB-attacked lodgepole pine, we could not separate out the growth release of other NE species like subalpine fir (p12, l8). What might be contentious is not the growth release itself, but the subset of our results suggesting this growth release might be strong enough to increase ecosystem carbon compared to the no-outbreak control. This possibility seems contradictory to previous modelling studies that quantified MPB impacts on ecosystem carbon or NEP... but these studies all

neglected the growth release of the non-target vegetation. To stress this point, we added the following sentence (p5, l16): "Previous modelling studies quantifying MPB impacts on $C_{eco}$ or NEP (Kurz et al., 2008; Edburg et al., 2011; Ghimire et al., 2015; Arora et al., 2016) have resorted to NEonly-type approaches that did not account for the possible growth release of the non-target vegetation". <<

*P14-L20: You could perhaps add a sentence explaining roughly what Kalb is representing.*

>> 3.10 Thank you for this suggestion. The text now reads (p14, l20; new text in blue): "To estimate $RF_{alb}$ *(m, y)*, we used the radiative kernels approach (Shell et al., 2008; Soden et al., 2008) which gives the radiative forcing caused by a unit change in a component of the climate system and has already been employed in previous studies on MPB-induced changes in $RF_{alb}$ (O'Halloran et al., 2012; Vanderhoof et al., 2014). Using the α radiative kernel for month *m* ($K_{alb}$ (*m*), in W m$^{-2}$), we could thus estimate the α-caused radiative forcing as [...]" (i.e., Kalb is the radiative kernel for albedo). <<

*Table 1: + symbols probably not necessary.*

>> 3.11 We removed the "+" symbols from Table 1. <<

*Correspondence to*: Jean-Sébastien Landry (jean-sebastien.landry2@mail.mcgill.ca)

[Figure]

**Figure S1.** Transient effect of the *Peak* outbreak regime on total *RF* as well as its $CO_2$ and $\alpha$ components (all in pico-W m$^{-2}$, for 1-ha outbreaks) compared with the no-outbreak *Control* (the outbreak occurred on year 1). The columns correspond to the three locations (Fig. 1) and the rows to the four vegetation coexistence scenarios (Table 2); the y-axis scale differs across the four rows.

**Table S1.** Comparison of IBIS–MIM results for different carbon cycle variables to four empirical and one model-based studies.

| Variable | This study | Previous studies |
|---|:---:|:---:|
| $\Delta C_{eco}$ $(\mathrm{kg\,C\,m^{-2}})$ | | |
| Years 1–3 | $-0.16$ $(-0.23$ to $-0.12)^a$ | $-1.9$ $(-4.2$ to $0.4)^1$ |
| $\Delta C_{eco}$ $(\mathrm{kg\,C\,m^{-2}})$ | | |
| Years 25–30 | $-0.20$ $(-0.58$ to $0.34)^b$ | $-0.40^2$ |
| $\Delta$GPP $(\mathrm{kg\,C\,m^{-2}\,yr^{-1}})$ | | |
| Years 1–3 | $-0.34$ $(-0.41$ to $-0.30)^a$ | $-0.04$ $(-0.09$ to $0.01)^3$ |
| Years 4–9 | $-0.24$ $(-0.50$ to $0.01)^a$ | $-0.22$ $(-0.27$ to $0.17)^3$ |
| $\Delta$GPP $(\%)$ | | |
| Years 1–5 | $-18$ $(-33$ to $-11)^c$ | $-14$ $(-18$ to $-10)^4$ |
| $\Delta$Ecosystem respiration $(\%)$ | | |
| Years 1–5 | $-9$ $(-16$ to $-5)^c$ | $-12$ $(-16$ to $-8)^4$ |
| $\Delta$NEP $(\mathrm{kg\,C\,m^{-2}\,yr^{-1}})$ | | |
| Years 1–4 | $-0.19$ $(-0.26$ to $-0.16)^a$; $-0.23^d$ | $-0.13^5$ |
| Years 5–15 | $-0.05$ $(-0.16$ to $0.09)^a$; $-0.15^d$ | $-0.23^5$ |
| Years 16–25 | $-0.01$ $(-0.07$ to $0.10)^a$; $-0.07^d$ | $-0.13^5$ |
| Years 26–65 | $0.00$ $(-0.04$ to $0.05)^a$; $-0.03^d$ | $-0.05^5$ |
| Years 66–80 | $0.02$ $(-0.01$ to $0.04)^a$; $0.03^d$ | $0.02^5$ |

[a]Mean value from the 12 combinations of locations and vegetation coexistence scenarios for the *Peak* outbreak regime, with the minimum and maximum values in parenthesis. [b]Same as "*a*", but for the *Medium* outbreak regime. [c]Same as "*a*", but for the *Large* outbreak regime. [d]Value from the *Peak* outbreak regime at the central location for the NEonly vegetation scenario.

[1]Morehouse et al. (2008), from their Table 3 ($\geq$80% mortality $\sim$1–3 years earlier); errors were added in quadrature.

[2]Kashian et al. (2013), from their Table 4 ($\sim$25% mortality 25–30 years earlier).

[3]Bright et al. (2013), from the >70–90% cumulative mortality of their Fig. 2 (timeseries over different stands); Year 0 corresponds to mortality of >1%.

[4]Moore et al. (2013), from their Fig. 1 (50% mortality at Year 0, 70% mortality at Year 5).

[5]Arora et al. (2016), from their Fig. S6 (estimated uncertainty of $\sim$0.03 $\mathrm{kg\,C\,m^{-2}\,yr^{-1}}$ for our visual retrieval of their results); $\sim$80% cumulative mortality over Years $\sim$1–10 for a NEonly-type modelling setting in a grid cell close to the central location of our study.

**Table S2.** Comparison of IBIS–MIM *Peak* results for $\Delta\alpha$ (unitless) to three empirical studies.

| Variable | This study[a,b] | Previous studies |
|---|---|---|
| Mean value; Nov–Feb | | |
| Years 4–9 | 0.07 (0.06 to 0.08) | 0.05 (0.02 to 0.08)[1] |
| Years 10–14 | 0.09 (0.08 to 0.10) | 0.07 (0.04 to 0.10)[1] |
| Mean value; Jun–Sep | | |
| Years 4–9 | 0.009 (−0.002 to 0.013) | 0.006[1] |
| Years 10–14 | 0.004 (−0.001 to 0.009) | 0.006[1] |
| February only | | |
| Years 1–3 | 0.011 (0.010 to 0.013) | 0.00 (−0.04 to 0.04)[2] |
| Years 4–9 | 0.09 (0.08 to 0.10) | 0.06 (−0.005 to 0.13)[2] |
| Mean value; Dec–Feb | | |
| Years 1–3 | 0.008 (0.007 to 0.009) | −0.02[3] |
| Years 4–13 | 0.07 (0.06 to 0.08) | 0.06[3] |
| Years 14–20 | 0.05 (0.04 to 0.07) | 0.10[3] |
| Years 21–30 | 0.11 (0.05 to 0.14) | 0.06[3] |
| Years 31–40 | 0.10 (0.03 to 0.16) | 0.00[3] |
| Years 51–60 | 0.05 (0.004 to 0.11) | −0.01[3] |
| Mean value; Jun–Aug | | |
| Years 1–3 | 0.001 (−0.001 to 0.002) | 0.005[3] |
| Years 4–13 | 0.007 (−0.001 to 0.01) | 0.005[3] |
| Years 14–20 | 0.003 (−0.001 to 0.006) | 0.005[3] |
| Years 21–30 | 0.009 (−0.002 to 0.02) | 0.005[3] |
| Years 31–40 | 0.008 (−0.002 to 0.02) | −0.005[3] |
| Years 51–60 | 0.004 (−0.001 to 0.01) | 0.01[3] |

[a] Mean value from the 12 combinations of locations and vegetation coexistence scenarios, with the minimum and maximum values in parenthesis. [b] Values for many months were computed as the simple mean of the monthly $\Delta\alpha$ (i.e., without weighting monhtly values with incoming solar radiation).

[1] O'Halloran et al. (2012), from the black curves and error bars of their Fig. 6 (timeseries over different stands); cumulative mortality and mortality at Year 1 not reported.

[2] Bright et al. (2013), from the >70–90% cumulative mortality of their Fig. 2 (timeseries over different stands); Year 0 corresponds to mortality of >1%.

[3] Vanderhoof et al. (2014), from the MODIS results of their Fig. 2 (stands grouped by age since mortality); cumulative mortality from <1 to 88%.

*Correspondence to:* Jean-Sébastien Landry (jean-sebastien.landry2@mail.mcgill.ca)

**Abstract.** The ongoing major outbreak of mountain pine beetle (MPB) in forests of western North America has led to considerable research efforts. Yet many questions remain unaddressed regarding its long-term impacts, especially when accounting for the range of possible responses from the non-target vegetation (i.e., deciduous trees and lower-canopy shrubs and grasses). We used the Integrated BIosphere Simulator (IBIS) process-based ecosystem model along with the recently incorporated

5   Marauding Insect Module (MIM) to quantify, over 240 years, the impacts of various MPB outbreak regimes on lodgepole pine merchantable biomass, ecosystem carbon, surface albedo, and the net radiative forcing on global climate caused by the changes in ecosystem carbon and albedo. We performed simulations for three locations in British Columbia, Canada, having different climatic conditions, and four scenarios of various coexisting vegetation types with variable growth release responses. The impacts of MPB outbreaks on merchantable biomass (decrease) and surface albedo (increase) were similar across the

10   12 combinations of locations and vegetation coexistence scenarios. The  impacts on ecosystem carbon and radiative forcing, however, varied substantially in magnitude and sign depending upon the presence and response of the non-target vegetation, particularly for the two locations not subjected to growing-season soil moisture stress; this variability represents the main finding from our study. Despite major uncertainty in the value of the resulting radiative forcing, a simple analysis also suggested that the MPB outbreak in British Columbia will have a smaller impact

15   on global temperature  over the coming decades and centuries than a single month of global anthropogenic $CO_2$ emissions from fossil fuel combustion and cement production. Moreover, we found that: (1) outbreak severity (i.e., per-event mortality) had a stronger effect than outbreak return interval on the variables studied, (2) MPB-induced changes in carbon dynamics had a stronger effect than concurrent changes in albedo on net radiative forcing, and (3) the physical presence of MPB-killed dead standing trees was potentially beneficial to tree regrowth. Given that

the variability of pre-outbreak vegetation characteristics can lead to very different regeneration pathways, the four vegetation coexistence scenarios we simulated probably only sampled the range of possible responses.

[revised manuscript text omitted]
  undergoes $100\%$ mortality in a grid cell (e.g., as occurred with our $Peak$ regime; see Section 2.3), this "seed" LAI can therefore initiate regeneration. IBIS does not simulate establishment of many individuals for the same PFT in a grid cell. Annual litterfall is divided into daily transfers to soil, where carbon decomposition is modelled as a function of microbial biomass, soil temperature, and moisture. IBIS results compare relatively well with empirical data over large regions and several field sites, including in Canada (Foley et al., 1996; Delire and Foley, 1999; Kucharik et al., 2000; Lenters et al., 2000; El Maayar et al., 2001, 2002; Kucharik et al., 2006).

MIM was designed to simulate the effects of insect outbreaks within process-based ecosystem models similar to IBIS (Landry et al., 2016). MIM prescribes, at a daily time step, the direct insect-caused vegetation damage (i.e., defoliation and/or mortality), an approach that is similar to the "pathogen and insect pathways" from Dietze and Matthes (2014). The resulting impacts on vegetation dynamics and land–atmosphere exchanges of carbon, energy, and water are estimated by the supporting ecosystem model as a function of the post-outbreak state of the vegetation. MIM currently represents the effects of vegetation damage caused by outbreaks of three insect functional types (IFTs): broadleaf defoliators, needleleaf defoliators, and bark beetles. The bark beetle IFT used here was parameterized based on MPB-caused mortality of lodgepole pine. When a MPB outbreak occurs, mortality is assumed to begin on 1 August and increases linearly over 50 days (Landry et al., 2016) until reaching the user-prescribed annual mortality level for the specific year (see Section 2.3). Killed trees become DSTs that interact with the exchanges of energy and water (e.g., absorbing solar radiation) but do not transpire or perform photosynthesis. DST carbon is gradually transferred to litter based on a pre-defined schedule: at year end for fine roots, over the three years following the year of mortality for needles, and, after a 5-year delay period, over 20 years for stem and coarse roots (Landry et al., 2016). IBIS then subdivides these annual amounts into daily transfers to soil.

IBIS–MIM results after a simulated MPB outbreak generally agreed with previous  studies (Landry et al., 2016). The only bias identified in that previous assessment of IBIS–MIM consisted of a lower snow depth/amount following MPB mortality vs. the no-outbreak control, contrary to the conclusion of most – but not all – previous studies. This bias likely resulted from overestimation by IBIS of heat storage within tree stems (Pollard and Thompson, 1995; El Maayar et al., 2001), including DSTs. IBIS–MIM might therefore underestimate the length of the snow cover season in MPB-attacked stands, thereby underestimating the consequent increases in springtime albedo and reflected solar radiation. This possible bias seems unlikely to be serious for the current study, because: (1) areal snow coverage, which matters more for albedo than snow depth/amount, was the same for the outbreak and control results during most of the snow cover season; and (2) the generally earlier and faster snowmelt caused by MPB outbreaks (Pugh and Small, 2012; Mikkelson et al., 2013) is consistent with IBIS–MIM results.

**2.3 Simulation design**

IBIS requires input data for soil texture and climatic variables related to temperature, humidity (including precipitation and cloud cover), and wind speed. For all variables, including the ones provided in Table 1, we used the same input data as Landry et al. (2016): the mean 1961–1990 atmospheric $CO_2$ level, gridded 1961–1990 monthly mean data for climate (New et al., 1999), and survey data from versions 2.1 and 2.2 of the Soil Landscapes of Canada for soil (http://sis.agr.gc.ca/cansis/nsdb/slc/index.html). Note that contrary to gap models, IBIS does not have an intrinsic spatial resolution; but as the computation of radiative forcing requires a specific area (see Appendix A), we used a nominal area of 1 ha here. Although we did not assess the effect of climate change, seeking to first understand the ecosystem responses within a stable climate regime, we considered the effect of varying climatic conditions by studying three locations in British Columbia, henceforth designated as northern, central, and southern (Fig. 1). These three locations have witnessed substantial MPB mortality since 2000 (Walton, 2013) under different climatic conditions (Table 1). The southern location is warmer than the central and northern locations. These last two locations have similar mean annual temperature, but the northern location has warmer summers and colder winters. Annual precipitation is similar in all locations, but summer rainfall is much lower in the southern location and results in drier conditions during the growing season.

We divided the simulations into four groups of different vegetation coexistence scenarios (Table 2). Five IBIS PFTs can coexist at the three locations considered here: the needleleaf evergreen (NE) trees targeted by MPB, broadleaf deciduous (BD) trees (e.g., trembling aspen; *Populus tremuloides* Michx.), and evergreen shrubs, cold-deciduous shrubs, and $C_3$ grasses in the lower canopy. The NEonly scenario allowed only NE trees and thus did not account for the possible response of the non-target vegetation. The NE-LC scenario allowed for the coexistence of NE trees and lower-canopy PFTs, but excluded BD trees. Note that in IBIS, coexisting PFTs compete for the same incoming solar radiation and soil water, instead of being segregated into independent sub-grid tiles as in many similar models, so that tree PFTs actually shade the underlying lower canopy. The NE-LCcons scenario was similar to NE-LC, except that the total biomass of the lower canopy was kept constant from the year of the first MPB outbreak onwards. Consequently, the lower canopy could increase its net primary productivity (NPP, in $kg\,C\,m^{-2}\,yr^{-1}$) following MPB mortality (e.g., due to higher light penetration), but the additional productivity was transferred to litterfall so that the lower canopy was unable to grow and expand, thereby preventing further increases in productivity. The purpose of this constraint was to account for the effect of vegetation growth limitations not included in IBIS, such as nutrient availability. Finally, the AllPFT scenario allowed the five PFTs to freely compete throughout all years. Previous modelling studies quantifying MPB impacts on $C_{eco}$ or NEP (Kurz et al., 2008; Edburg et al., 2011; Ghimire et al., 2015; Arora et al., 2016) have resorted to NEonly-type approaches that did not account for the possible growth release of the non-target vegetation.

[revised manuscript text omitted]

we identified among all our $Peak$ simulations the two instances that led to the most extreme (positive and negative) annual $RF$ values. We then recomputed these $RF$ trajectories for an area of 18.1 Mha, which is the total area affected by the MPB outbreak (British Columbia, 2012b). Finally, we determined the value of a single pulse of actual (positive $RF$) or avoided (negative $RF$) fossil fuel $CO_2$ emissions that would invariably have, throughout the 240 years, a stronger radiative forcing than the MPB-caused $RF$ (see Appendix A). This approach likely overestimated the maximum annual impact of MPB on the global climate for three reasons. First, not all area affected by MPB suffered 100 % mortality as prescribed in $Peak$. Second, the single-year $Peak$ mortality led to stronger extreme $RF$ values compared to a gradual increase and decrease of the outbreak over many years. Third, the amount and composition of non-target vegetation are highly variable among MPB-attacked stands (Axelson et al., 2009; Amoroso et al., 2013; Hawkins et al., 2013; Pelz and Smith, 2013; Alfaro et al., 2015; Campbell and Antos, 2015) and this variability appears consequential for $RF$ (see Section 3); hence the $RF$ response from the same vegetation coexistence scenario was unlikely representative of the mean response over the entire area affected.

**3 Results**

**3.1 Comparison to previous studies**

IBIS–MIM results following a MPB outbreak have already been found to agree with results from 38 field-, satellite-, and model-based studies for 28 variables related to vegetation dynamics and the exchanges of carbon, energy, and water, over daily to multi-annual timescales (Landry et al., 2016) . That previous assessment also provided time-since-disturbance NPP results for different PFTs under the NE-LCcons scenario. Here we performed an additional assessment of IBIS–MIM, comparing its results for different variables related to the carbon cycle and for $\Delta\alpha$ to previous studies (see Tables S1-S2 in the Supplement). We restricted these comparisons to empirical studies that included control stands, except for the modelling study of Arora et al. (2016) that used a model similar to IBIS–MIM under a NEonly-type setting and provided results for one grid cell close to our central location. Given the differences in locations, cumulative mortality levels, and temporal patterns of mortality, IBIS–MIM agreed reasonably well with previous studies, providing a measure of confidence in the realism of the results shown below.

**3.2 Transient results**

MPB-caused reductions in $B_{merch}$ were similar across the three locations and four vegetation coexistence scenarios (Fig. 2). For the $Peak$ regime, the single 100 % mortality event removed all $B_{merch}$, after which 20–50 years were needed before NE trees became big enough to have any commercial value. The slower recovery of $B_{merch}$ in AllPFT compared to other scenarios resulted from the growth release of BD trees, which were able to grow strongly for a few decades but were very poor long-term competitors at all locations. For the three periodic regimes, the recurring MPB outbreaks prevented $B_{merch}$ from recovering to the $Control$ value as in $Peak$. We found some evidence of a biophysical interaction between DSTs and regrowing NE trees for the $Peak$ regime in the NEonly, NE-LCcons, and NE-LC scenarios: after the 100 % mortality event, NPP of NE trees increased rapidly but, after about 20–25 years, decreased noticeably for ~10 years before resuming again (similarly to

the results shown in Fig. 2 from Landry et al., 2016). We believe this response resulted from the interception of radiation by DSTs, which warmed the upper canopy and initially allowed regrowing NE trees to perform photosynthesis more efficiently at a higher temperature. This warming effect decreased as DSTs fell, causing productivity of the regrowing NE trees to decline even though they were intercepting more light. In the AllPFT scenario, the slower regrowth of NE trees caused them to miss the warming effect due to the presence of DSTs.

The impacts of MPB on $C_{eco}$ (Fig. 3) were much more complex than on $B_{merch}$. In the NEonly scenario, the three periodic regimes led to gradual declines in $C_{eco}$ that showed indications of possible stabilization towards the end of the simulations, whereas for the $Peak$ regime, $C_{eco}$ was still recovering after 240 years. The results were qualitatively the same in the NE-LCcons scenario, albeit with much smaller $\Delta C_{eco}$ because the lower-canopy growth release partially compensated for the death of NE trees. At the southern location, the growing-season soil moisture stress probably explains why the growth release of non-target PFTs was only marginally stronger in the unconstrained NE-LC and AllPFT scenarios vs. NE-LCcons. Conversely, MPB outbreaks substantially increased total NPP at the northern and central locations for NE-LC and AllPFT, by inducing a strong growth release of the non-target vegetation and fostering the increased coexistence of PFTs occupying different ecological niches (upper vs. lower canopies, and evergreen needleleaf vs. deciduous broadleaf strategies) compared to undisturbed forests dominated by lodgepole pine. Here, the higher productivity of the non-target vegetation exceeded the productivity losses and gradual decomposition of killed NE trees; hence, after a delay of a few years to a few decades, $\Delta C_{eco}$ switched from negative to positive (Fig. 3, panels g, h, j, and k).

MPB outbreaks increased $\alpha$ for all locations and vegetation coexistence scenarios (Fig. 4). Results were very similar across locations and scenarios, except for smaller $\alpha$ increases in AllPFT (panels j, k, and l) for the $Peak$ regime due to the absorption of radiation by BD trees (even when leafless during winter, as IBIS accounts for the snow-masking effect of stems) following their growth release.

$RF$ varied substantially across locations and vegetation coexistence scenarios (Fig. 5; also see Fig. S1 in the Supplement for the $CO_2$-based and $\alpha$-based components of the total $RF$ for $Peak$). 
[revised manuscript text omitted]
., 2007). MPB outbreaks can also affect the global climate through other mechanisms that are poorly constrained and that we did not consider, for example changes in atmospheric chemistry (Arneth and Niinemets, 2010) or in the partitioning between latent and sensible heat fluxes (Ban-Weiss et al., 2011).

**5 Conclusions**

Despite major progress over the last decades, various knowledge gaps still limit the understanding of the consequences of mountain pine beetle (MPB) outbreaks. In this study, we used a climate-driven process-based ecosystem model to estimate the  long-term  impacts of prescribed MPB outbreaks.

We found that the differences in vegetation coexistence scenario and location had little influence over MPB impacts on lodgepole pine merchantable biomass ($B_{merch}$; Fig. 2) and surface albedo ($\alpha$; Fig. 4). On the contrary, accounting for the non-target vegetation invariably reduced losses of ecosystem carbon ($C_{eco}$) and, at the two locations not subjected to growing-season soil moisture stress, even led to post-outbreak $C_{eco}$ increases for the two vegetation coexistence scenarios with the strongest growth release  (Fig. 3). Although MPB-induced increases in $\alpha$ always had a cooling influence, the net global warming or cooling effect of MPB outbreaks was determined by the  stronger carbon-based responses (see Fig. 6, and compare Fig. 5 with Fig. 3). A simple analysis suggested that the MPB outbreak in British

Columbia will have less influence on global temperature over the coming centuries than one month of global anthropogenic $CO_2$ emissions at the 2014 level (Fig. 7).

5     The management and research implications of our study are fourfold. First, salvage logging, particularly when performed as clear-cut harvesting, may be detrimental to carbon stewardship when surviving trees and other lower-canopy vegetation are abundant. Second, salvage logging could slow forest recovery if, following high MPB mortality, tree productivity is indeed increased due to the physical presence of dead standing trees, a hypothesis that should be subject to empirical studies. Third, MPB disturbances might not necessarily lead to global warming, so activities aiming to prevent or control outbreaks

10 (e.g., pre-emptive logging) should not be heralded as climate mitigation strategies without more detailed analyses. Fourth, the substantial spatiotemporal variability in MPB-induced changes suggests a need to support field studies that encompass a wide range of stand conditions and are maintained over several decades.

**Code availability**

IBIS–MIM code is available upon request from the corresponding author or through the following link: http://landuse.geog.

15 mcgill.ca/~jean-sebastien.landry2@mail.mcgill.ca/ibismim/.

**Data availability**

Simulation results (as NetCDF files; http://www.unidata.ucar.edu/software/netcdf/) are available upon request from the corresponding author.

**Appendix A: Additional computations**

20 **A1   Merchantable biomass**

$B_{merch}$ is a fraction of the total tree biomass in a forest, because immature trees, as well as tops branches, and stumps from mature trees, are excluded. We computed $B_{merch}$ as the product of $prop$, which is the proportion (unitless) of the total tree biomass that is merchantable, and $B_{tot}$, which is the total tree biomass (in $\mathrm{kg\,C\,m^{-2}}$) estimated by IBIS. We derived $prop$ from Figure 5 of Kurz et al. (2009), which shows $B_{merch}$ and $B_{tot}$ as a function of time for a lodgepole pine stand:

25 $$prop = \begin{cases} 0 & \text{if } B_{tot}/B_{max} < 0.21 \\ 0.5058 + 0.3172 \times \ln\left(B_{tot}/B_{max}\right) & \text{otherwise} \end{cases} \tag{A1}$$

where $B_{max}$ is the maximum tree biomass (in kg C m$^{-2}$) at equilibrium. The logarithmic function for *prop* provided a good fit ($R^2 = 0.996$) with the data extracted from Kurz et al. (2009).

**A2 Radiative forcing**

**A2.1 Mountain pine beetle**

5    The net $RF$ was the sum of the radiative forcing from atmospheric $CO_2$ changes ($RF_{CO_2}$) and from $\alpha$ changes ($RF_{alb}$). $RF_{CO_2}$ in year $y$ caused by a change in atmospheric $CO_2$ in the same year ($\Delta C(y)$, in ppmv) compared to a reference concentration ($C_o$, in ppmv) was given by (Myhre et al., 1998):

$$RF_{CO_2}(y) = 5.35 \times \ln\left(1 + \frac{\Delta C(y)}{C_o}\right) \tag{A2}$$

The change in atmospheric $CO_2$ in year $y$ due to MPB outbreaks resulted from all past changes in $C_{eco}$ computed by IBIS–
10    MIM, including vegetation regrowth, while accounting for the airborne fraction of these past fluxes. In other words, $\Delta C(y)$ was the convolution of the series of past yearly land-to-atmosphere fluxes with the impulse response function ($IRF$) for the airborne fraction of these past fluxes. Since $\Delta C(y)$ is an absolute amount and not a change per unit of land area disturbed, we must compute it for a specific area $A_{MPB}$ (in m$^2$) of MPB mortality. We thus computed $\Delta C(y)$ as:

$$\Delta C(y) = A_{MPB} \times k \times \sum_{t=0}^{y-1} \delta C_{eco}(y-t) \times IRF(t) \tag{A3}$$

15    where $k$ is equal to $4.69 \times 10^{-13}$ ppmv per kg C (CDIAC, 2012) and $\delta C_{eco}(x) = (C_{eco,control}(x) - C_{eco,control}(x-1)) - (C_{eco,MPB}(x) - C_{eco,MPB}(x-1))$. For the $IRF(t)$ function (unitless), we used the mean response from the injection of a single pulse of $CO_2$ into 15 different coupled climate–carbon models (Joos et al., 2013). A similar approach has already been used to estimate $RF_{CO_2}$ from MPB outbreaks (O'Halloran et al., 2012).

We estimated the radiative forcing in year $y$ caused by a change in $\alpha$ as the mean of monthly values:

20    $$RF_{alb}(y) = \frac{1}{365} \times \sum_{m=1}^{12} n_{days}(m) \times RF_{alb}(m, y) \tag{A4}$$

where $n_{days}(m)$ is the number of days in month $m$, and $RF_{alb}(m, y)$ is the average $\alpha$-caused radiative forcing in month $m$ of year $y$. To estimate $RF_{alb}(m, y)$, we used the radiative kernels approach (Shell et al., 2008; Soden et al., 2008) which gives the radiative forcing caused by a unit change in a component of the climate system and has already been employed in previous studies on MPB-induced changes in $RF_{alb}$ (O'Halloran et al., 2012; Vanderhoof et al., 2014). Using the $\alpha$ radiative

kernel for month $m$ ($K_{alb}(m)$, in $\mathrm{W\,m^{-2}}$), we could thus estimate the $\alpha$-caused radiative forcing as (Shell et al., 2008; Soden et al., 2008):

$$RF_{alb}(m,y) = \frac{A_{MPB}}{A_{Earth}} \times K_{alb}(m) \times \Delta\alpha(m,y) \tag{A5}$$

where $A_{Earth}$ is the Earth area ($5.1 \times 10^{14}\,\mathrm{m^2}$; Wallace and Hobbs (2006)) and $\Delta\alpha(m,y)$ is the change in $\alpha$ between a simulation with MPB mortality and the $Control$ simulation. We averaged $K_{alb}(m)$ from two models: the Community Atmospheric Model (data downloaded from http://people.oregonstate.edu/~shellk/kernel.html) and the Geophysical Fluid Dynamics Laboratory atmospheric model (data downloaded from http://www.rsmas.miami.edu/personal/bsoden/data/kernels.html).

We computed the net $RF$ for an outbreak area of 1 ha (i.e., $A_{MPB} = 10{,}000\,\mathrm{m^2}$). Strictly speaking, the $RF$ results we obtained cannot be directly scaled as a function of the disturbed area, because Eq. (A2) is not linear and the $K_{alb}(m)$ values in Eq. (A5) vary spatially. Yet both these restrictions can be ignored for regional-level scaling of our $RF$ results. First, the changes in total atmospheric $CO_2$ were very small, so that Eq. (A2) varied almost linearly with $\Delta C(y)$ as $\ln(1+x) \approx x$ for very small values of $x$. Second, $K_{alb}(m)$ values were computed at a coarse resolution of $\geq 2°$ and are spatially correlated. Therefore, $RF$ for a MPB-disturbed area of 10,000 ha, for example, would be very well approximated by multiplying our reported results by a factor of 10,000.

**A2.2 Fossil fuel CO$_2$**

The radiative forcing caused by a positive or negative pulse of fossil fuel $CO_2$ emissions was also computed based on Eq. (A2), with $\Delta C(y)$ being given by:

$$\Delta C(y) = k \times P \times IRF(y-1) \tag{A6}$$

where $P$ is the value of the single pulse of emissions (in kg C) occurring on year 1, with $k$ and $IRF$ as in Eq. (A3). We varied $P$ until obtaining a radiative forcing response that was invariably greater (smaller) than the bounding MPB-caused positive (negative) $RF$ response throughout the 240 years.

*Author contributions.* J.-S. Landry designed the study with advice from L. Parrott, D. T. Price, and N. Ramankutty; J.-S. Landry performed the simulations with IBIS–MIM and analyzed the results; J.-S. Landry prepared the manuscript with contributions from all co-authors.

*Acknowledgements.* We thank Dany Plouffe for producing Fig. 1 and the Concordia Climate Lab for feedback on the results. The comments from three reviewers helped us improve the manuscript. J.-S. Landry was funded by a doctoral scholarship (B2) from the Fonds de recherche du Québec – Nature et technologies (FRQNT).

[revised manuscript text omitted]